# On the suitability of the Thorpe-Mason model for Calculating Sublimation of Saltating Snow

Varun Sharma[1], Francesco Comola[1], and Michael Lehning[1,2]

[1]School of Architecture, Civil and Environmental Engineering, Swiss Federal Institute of Technology, Lausanne, Switzerland
[2]WSL Institute for Snow and Avalanche Research SLF, Davos, Switzerland

**Correspondence:** Varun Sharma (varun.sharma@epfl.ch)

**Abstract.** The Thorpe and Mason (TM) model for calculating the mass lost from a sublimating snow grain is the basis of all existing small and large-scale estimates of drifting snow sublimation and the associated snow mass balance of polar and alpine regions. We revisit this model to test its validity for calculating sublimation from saltating snow grains. It is shown that numerical solutions of the unsteady mass and heat balance equations of an individual snow grain reconcile well with the steady-state solution of the TM model, albeit after a transient regime. Using large-eddy simulations (LES), it is found that the residence time of a typical saltating particle is shorter than the period of the transient regime, implying that using the steady state solution might be erroneous. For scenarios with equal initial air and particle temperatures of 263.15 K, these errors range from 26% for low-wind low saturation-rate conditions to 38% for high-wind high saturation-rate conditions. With a small temperature difference of 1 K between the air and the snow particles, the errors due to the TM model are already as high as 100% with errors increasing for larger temperature differences.

## 1  Introduction

Sublimation of drifting and blowing snow has been recognized as an important component of the mass budget of polar and alpine regions (Liston and Sturm, 2004; van den Broeke et al., 2006; Lenaerts et al., 2012; Vionnet et al., 2014). Field observations and modeling efforts focused on Antarctica have highlighted the fact that precipitation and sublimation losses are the dominant terms of the mass budget in the katabatic flow region as well as the coastal plains (van den Broeke et al., 2006). Even though precipitation is challenging to measure accurately, methods to measure it exist, for example, using radar (Grazioli et al., 2017) or snow depth change (Vögeli et al., 2016). In comparison, sublimation losses are even harder to measure and can only be calculated implicitly; using measurements of wind speed, temperature and humidity. Thus, in regions where sublimation loss is a dominant term of the mass balance, it is also a major source of error. This error ultimately results in errors in the mass accumulation of ice on Antarctica, which is a crucial quantity for understanding sea-level rise and climate change (Rémy and Frezzotti, 2006; Rignot et al., 2011; Lenaerts et al., 2012).

Aeolian transport of snow can be classified into three modes, namely, creeping, saltation and suspension. Creeping consists of heavy particles rolling and sliding along the surface of the snowpack either due to form drag or bombardment due to impacting particles. Saltation consists of particles being transported along the surface via short, ballistic trajectories with heights mostly less than $10\,\mathrm{cm}$ and involves mechanisms of aerodynamic entrainment along with rebound and splashing of ice grains (Doorschot and Lehning, 2002; Comola and Lehning, 2017). Suspension on the other hand refers to transport of small ice grains at higher elevations and over large distances without contact with the surface. Current calculations of sublimation losses are largely restricted to losses from ice grains in suspension. This is true for both field studies (Mann et al., 2000), where sublimation losses are calculated using measurements, usually at the height of $\mathcal{O}(1\,\mathrm{m})$, and in mesoscale modeling studies (Xiao et al., 2000; Déry and Yau, 2002; Groot Zwaaftink et al., 2011; Vionnet et al., 2014), where the computational grids and time-steps are too large to resolve flow dynamics at saltation length and time scales. Mass loss in the saltation layer is hard to measure and is neglected based on the justification that the saltation layer is saturated. However, recent studies using high-resolution steady-flow, Reynolds-averaged Navier-Stokes (RANS) type simulations (Dai and Huang, 2014) claim that sublimation losses in the saltation layer are not negligible, particularly for wind speeds close to the threshold velocities for aeolian transport, wherein a majority of aeolian snow transport occurs via saltation rather than suspension.

The coupled heat and mass balance equations of a single ice particle immersed in turbulent flow are

$$c_i\, m_p \frac{\mathrm{d}T_p}{\mathrm{d}t} = L_s \frac{\mathrm{d}m_p}{\mathrm{d}t} + \pi \mathcal{K}\, d_p\, (T_{a,\infty} - T_p)\mathcal{N}u, \tag{1}$$

$$\frac{\mathrm{d}m_p}{\mathrm{d}t} = \pi \mathcal{D} d_p\, (\rho_{w,\infty} - \rho_{w,p})\,\mathcal{S}h, \tag{2}$$

where, $m_p$, $T_p$ and $d_p$ are the mass, temperature and diameter of the particle respectively that vary with time, $c_i$ is the specific heat capacity of ice, $L_s$ is the latent heat of sublimation, $\mathcal{K}$ is the thermal conductivity of moist air and $\mathcal{D}$ is the mass diffusivity of water vapor in air. Transfer of heat and mass is driven by differences of temperature and vapor density between the particle surface ($T_p$, $\rho_{w,p}$) and the surrounding fluid ($T_{a,\infty}$, $\rho_{w,\infty}$). The vapor density at the surface of the ice particle is considered to be the saturation vapor density for the particle temperature. The transfer mechanisms are enhanced by turbulence, the effect of which is parameterized by the Nusselt ($\mathcal{N}u$) and Sherwood ($\mathcal{S}h$) numbers respectively. $\mathcal{N}u$ and $\mathcal{S}h$ are related to the relative speed ($|\boldsymbol{u}_{rel}|$) between the air and the particle via the particle Reynolds number ($Re_p$) as

$$Re_p = \frac{d\,|\boldsymbol{u}_{rel}|}{\nu_{air}}\,;\, \mathcal{N}u = 1.79 + 0.606\, Re_p^{1/2} Pr^{1/3}\,;\, \mathcal{S}h = 1.79 + 0.606\, Re_p^{1/2} Sc^{1/3}\,, \tag{3}$$

where $\nu_{air}$ is the kinematic viscosity of air and $Pr$ and $Sc$ are the Prandtl and Schmidt numbers respectively.

Thorpe and Mason (1966) solved the above coupled Eqs. (1) and (2) by, (a) neglecting the thermal inertia of the ice particle, thus effectively stating that all the heat necessary for sublimation is supplied by the air, and (b) considering the temperature difference between the particle and surrounding air to be *small*, thereby allowing for Taylor series expansion of the Clausius-Clapeyron equation and neglecting higher-order terms, resulting in their formulation for the mass loss term as,

$$\frac{\mathrm{d}m_p}{\mathrm{d}t} = \pi d_p\,(\sigma_* - 1)\left/\left(\frac{L_s}{\mathcal{K} T_{a,\infty} \mathcal{N}u}\left(\frac{L_s M}{R T_{a,\infty}} - 1\right) + \frac{1}{\mathcal{D}\rho_s(T_{a,\infty})\mathcal{S}h}\right)\right., \tag{4}$$

where $\rho_s(T_{a,\infty})$ is the saturation vapor density of air surrounding the particle, saturation-rate $\sigma_* = \rho_{w,\infty}/\rho_s(T_{a,\infty})$, $M$ is the molecular weight of water and $R$ is the universal gas constant. The above formulation has been used extensively to analyze data collected in the field (Mann et al., 2000), wind tunnel experiments (Wever et al., 2009), and numerical simulations of drifting and blowing snow (Déry and Yau, 2002; Groot Zwaaftink et al., 2011; Vionnet et al., 2014). In the modeling studies, the mass loss term is computed using Eq. (4) and is added, with proper normalization, to the advection-diffusion equation of specific humidity while the latent heat of sublimation multiplied by the mass loss term is added to the corresponding equation for temperature (Groot Zwaaftink et al., 2011).

Two observations motivated us to investigate the suitability of the TM model for sublimation of saltating snow particles. Firstly, the TM model assumes that all the energy required for sublimation is supplied by the air. This assumption was tested by Dover (1993) who compared the potential rates of cooling of particles with that of the surrounding air due to sublimation. Using scale analysis, Dover (1993) formulated the quantity $\xi = 6\rho_{air}c_{p,air} / \pi\rho_i c_i \overline{d_p}^3 N$, where $\overline{d_p}$ is the mean particle diameter, $N$ is the particle number density, $\rho_i$ is the density of ice, and showed that for $\xi \gg 1$, it can be accurately considered that the heat necessary for sublimation comes from the air. For standard values for an ice particle in suspension, $\overline{d_p} = 50\,\mu\mathrm{m}$ and $N \sim \mathcal{O}(10^6)$, this condition is easily met $\left(\xi \sim \mathcal{O}(10^3)\right)$. However, if we input values typical for saltation, i.e, $\overline{d_p} = 200\,\mu\mathrm{m}$ and $N \sim \mathcal{O}(10^8)$, $\xi \sim \mathcal{O}(1)$, and the condition is not met. Thus, for sublimation of saltating particles, it is important to consider the thermal inertia of the particles. A similar conclusion was reached in other modeling studies on topics of heat and mass exchange between disperse particulate matter in turbulent flow such as small water droplets in heat exchangers (Russo et al., 2014) and sea-sprays (Helgans and Richter, 2016).

Secondly, Eq. (4) computes mass loss as being directly proportional to $\sigma_*$ and neglects the temperature difference between the particle and air. Eq. (4) thus predicts a mass loss even in extremely high saturation-rate conditions, whereas immediate deposition of water vapor would occur on a particle even slightly colder than the air. Indeed, some field experiments have reported deposition as opposed to sublimation which was expected, on the basis of the measured under-saturation of the environment, particularly near coastal polar regions (Sturm et al., 2002). A simple everyday observation illustrates this fact clearly; There is immediate deposition of vapor and formation of small droplets on the surface of a cold bottle of beer even in room conditions with moderate humidity!

Motivated by the observations described above, in this article, we describe four numerical experiments where we compare differences between the fully numerical and the Thorpe and Mason (1966) solutions (referred to as NUM and TM approaches respectively). In Experiment I and II, we numerically solve Eqs. (1) and (2) and compare the results with Eq. (4) for physically plausible values of a saltating ice particle. Results of these tests are presented in Sect. 2. High-resolution large-eddy simulations (LES) of the atmospheric surface layer with saltating snow are performed for a range of environmental conditions to compute the differences between the NUM and TM approaches in realistic wind-driven saltating events. These results are presented in Sect. 3. A summary of the article is made in Sect. 4

## 2 Comparison between NUM and TM solutions: EXPERIMENT I and II

We consider an idealized scenario where a solitary spherical ice particle is held still in a turbulent air flow with constant mean speed, temperature and under-saturation. The evolution of the mass, diameter and temperature of the ice particle is calculated using both the NUM and TM models and an inter-comparison is made. This scenario is similar to the wind-tunnel study performed by Thorpe and Mason (1966) who measured mass loss of solitary ice grains suspended on fine fibers. In this scenario, we consider that the heat and mass transfer between the ice particle and the air changes the mass and temperature of the particle only while the mass and energy anomalies in the air are rapidly advected and mixed away. This implies that the environmental conditions subjected to the ice particle remain constant. While it can be expected that the environmental conditions will vary along the trajectory of a ice particle undergoing saltation or suspension, it is nevertheless useful to perform this analysis as it reveals important characteristics about the heat and mass evolution of a ice particle during sublimation and about the approximations used to derive the TM model.

For the NUM approach, Eqs. (1) and (2) are solved in a coupled manner using a simple first-order finite-differencing scheme for time-stepping with a time-step of $50\,\mu$s. For the TM approach, Eq. (4) is used with a similar numerical setup as for the NUM approach. In the TM approach, particle temperature is not considered and the mass and energy transfer is determined only by air temperature and saturation-rate. The initial particle diameter ($d_{p,IC}$) is $200\,\mu$m and the air-flow temperature is $263.15\,$K for both the NUM and TM approaches. We use a constant air speed of $5\,\text{ms}^{-1}$ resulting in $Re_p = 80$, $\mathcal{N}u = 6.7$ and $\mathcal{S}h = 6.5$ (using Eq. (3)). The values used here are typical of a saltating ice particle (Thorpe and Mason, 1966; Kok and Renno, 2009; Vionnet et al., 2014).

In Experiment I, we study the heat and mass output from a sublimating ice grain as a function of time. In the first case, Experiment I-A, we consider the effect of three different values of air-flow saturation-rate ($\sigma_* = 0.8$, 0.9 and 0.95) on differences between NUM and TM solutions. The NUM approach requires specification of the initial condition for the particle temperature ($T_{p,IC}$). In Experiment I-A, ($T_{p,IC}$) is taken to be the same as the air-flow temperature for the NUM approach, i.e, 263.15 K. Results for Experiment I-A are shown in Fig. 1(a-c), with subfigure (a) showing the mass output rate, $F_M$ and subfigure (b) showing the heat output rate, $F_Q$. Note that in this figure and subsequent figures, +(-) signifies mass and heat gained (lost) by the air. Since we keep the temperature and under-saturation of the air constant, the solutions of the TM approach are "steady-state" solutions with constant heat and mass transfer rates as seen in Fig. 1a and b. On the other hand, since the NUM approach solves the coupled equations that consider the evolution of the particle temperature, the heat and mass transfer rates evolve with time.

It can be seen that the NUM solutions initially evolve with time and reconcile with the steady-state TM solutions after a transient regime of about 0.3 seconds. Since the initial temperature of the particle is the same as the air, there is no heat transfer between the air and the particle (see the second term of the R.H.S of Eq. 1) initially. Thus, all heat transfer rates are initially zero for the NUM case in Fig. 1b. The under-saturation of the air forces mass transfer from the ice particle to air and the energy for the phase change comes from the internal energy of the ice particle. This causes the particle temperature to drop (see Fig. 2 below). With the particle now colder than the air, heat transfer from the air to the particle commences and ultimately, the

energy for sublimation comes entirely from the heat extracted from the air. The initial dynamics of the heat and mass transfer are completely neglected by the TM approach. In subfigure (c), the errors $\left(Err(t) = \left(\int_0^t F_{NUM}dt \middle/ \int_0^t F_{TM}dt - 1\right) * 100\right)$ for mass, $Err_M$ and heat, $Err_Q$ are shown. The errors reduce dramatically with time (for example, 15% at 0.3 seconds) and interestingly do not depend on the saturation-rate of the air-flow.

In the following case, Experiment I-B, similar simulations as in Experiment I-A are performed, but with $\sigma_* = 0.95$ while the initial temperature difference between the particle and the air is varied as $T_{p,IC} - T_{Air} = -2, -1, 1, 2\,\mathrm{K}$. The results are shown in Fig. 1(d-f). It is interesting to note that for each of the four cases considered, the TM solution predicts sublimation of the particle (consistent with $\sigma_* < 1$, see numerator of R.H.S of Eq. 3). On the other hand, for cases with colder particles, the NUM solutions show that there is initially deposition on the particle, along with larger values of heat absorbed from the

air. Correspondingly, in the cases with particles being warmer than the air, the mass loss is much higher in the NUM solution than that computed by the TM solution while the heat gained by the particle is also much higher. These higher differences are reflected in the $Err_M$ and $Err_Q$ curves in subfigure (f) where errors are found to be an order of magnitude higher that those in subfigure (c).

     We define relaxation time ($\tau_{relaxation}$) as the time required for the NUM solution to reconcile with the TM solution. The

importance of this quantity lies in the fact that if the residence time of a saltating ice grain in air is shorter than $\tau_{relaxation}$, the TM approach is likely to be erroneous and the NUM approach would be required. It is intuitive that $\tau_{relaxation}$ increases with $d_p$ on account of increasing inertia and decreases with $|\boldsymbol{u}_{rel}|$ due to more vigorous heat and mass transfer. Experiment I was repeated for values of $d_p$ and $|\boldsymbol{u}_{rel}|$ ranging between $(50 - 1000\,\mu\mathrm{m})$ and $\left(0 - 10\,\mathrm{ms}^{-1}\right)$ respectively. The upper-bound of the wind-speed range is quite high and it is extremely unlikely to find $|\boldsymbol{u}_{rel}| > 10\,\mathrm{ms}^{-1}$ in naturally-occurring aeolian transport.

Numerical results indeed confirm our intuition and it is found that for any given value of $|\boldsymbol{u}_{rel}|$, $\tau_{relaxation}$ is found to be $\propto d_p^\alpha$, where $\alpha\,(\sim 1.65)$. Furthermore, $\tau_{relaxation}$ decreases monotonically with increasing $|\boldsymbol{u}_{rel}|$ for a given value of $d_p$. For $d_p = 200\,\mu\mathrm{m}$, the plausible values of $\tau_{relaxation}$ are found to lie between 0.28 and 1.5 seconds (for $|\boldsymbol{u}_{rel}| = 10$ and $0\ \mathrm{ms}^{-1}$ respectively). Interestingly, $\tau_{relaxation}$ is not found to depend on either the saturation-rate of air or the difference between the initial particle and air temperature. Plots of $\tau_{relaxation}$ are highly relevant to discussion in Sect. 3 and presented there.

In Fig. 2, evolution of particle diameter ($d_p$) and temperature ($T_p$) is presented with subfigures (a) and (b) respectively describing the evolution for simulations in Experiment I-A with (c) and (d) being the corresponding results from Experiment I-B. In Experiment I-A, the particle diameters reduce linearly with time for both the NUM and TM approaches with the more shrinking (or in other words, sublimation) in the NUM solutions. More interesting is the evolution of the particle temperature, where the particle undergoes significant cooling due to sublimation and ultimately achieves a constant temperature. For exam-

ple, in the case for $\sigma_* = 0.8$, the particle temperature is ultimately 0.85 K lower than the initial particle temperature of 263.15 K. Note that for the TM approach, particle temperature is of no consequence and it is shown simply for reference.

     Following results of Experiment I, in Experiment II, we explore the parameter space of $(\sigma_*,\ T_{p,IC} - T_{Air})$ and compute the total mass ($M = \int_0^t F_M dt$) and total heat ($Q = \int_0^t F_Q dt$) output by a sublimating ice grain for a finite time of $t = 0.5$ seconds. Results shown in Fig. 3 subfigures (a) and (b) provide a comparison of the total mass lost using the NUM and TM solutions

respectively and the corresponding error is shown in subfigure (c). Similar figures are presented for the total heat lost/gained

by the air in subfigures (d-f). The inclusion of the inertial terms essentially causes the contours to be sloped for the NUM solution while the TM solutions do not depend on $T_{p,IC} - T_{Air}$ as expected. The error between the NUM and TM solutions are accentuated at high saturation-rate regimes, with errors larger than 30 % for $\sigma_* > 0.8$.

In summary, Experiments I and II highlight the fact that during the sublimation of an ice grain, there exists a finite, well-defined transient regime before the NUM solutions match the steady-state TM solutions. Furthermore, the NUM and TM solutions diverge rapidly with slight temperature differences between the particle and the air and with increasing $\sigma_*$ (which is a cause of concern since in snow-covered environments, the air usually is highly saturated). The results described above prompt an interesting question: are the residence times of saltating ice particles comparable to $\tau_{relaxation}$ ? We use large-eddy simulations to answer this question in the following section.

## 3 Large-eddy Simulations of Saltating Snow

### 3.1 Experiment III and IV: Simulation Details

To further understand the implications of the differences between the NUM and the TM approach, we performed LES of the atmospheric surface layer with an erodible snow surface as the lower boundary. We describe here only the main details of the LES that are relevant to our discussion with full model description along with equations presented in Supplementary Material S1. The LES solves filtered Eulerian equations for momentum, temperature and specific humidity on a computational domain of $6.4\,\mathrm{m} \times 6.4\,\mathrm{m}$ in the horizontal directions with vertical extent of the domain being $6.4\,\mathrm{m}$ as well. The snow surface, which constitutes the lower boundary of the computational domain, consists of spherical snow particles with a log-normal size distribution with a mean particle diameter of $200\ \mu\mathrm{m}$ and standard-deviation of $100\ \mu\mathrm{m}$. The coupling between the erodible snow-bed and the atmosphere is modeled through statistical models for aerodynamic entrainment (Anderson and Haff, 1988), splashing and rebounding of particle grains (Kok and Renno, 2009), which have been updated recently by Comola and Lehning (2017) to include the effects of cohesion and heterogeneous particle sizes. The use of these models essentially allows for overcoming the immense computational cost of resolving individual grain-to-grain interactions and allow us to consider the snow-surface as a *bulk* quantity rather than a collection of millions of individual snow particles. Once the ice grains are in the flow, their equations of motion are solved in the Lagrangian frame of reference with only gravitational and turbulent form drag forces included. Since the particle velocities are known, $|\boldsymbol{u}_{rel}|$ is calculated explicitly and used to compute $Re_p$, $\mathcal{N}u$ and $\mathcal{S}h$. The horizontal boundaries of the domain are periodic and the lower boundary condition (LBC) for velocity uses flux parameterizations based on Monin-Obukhov similarity theory, additionally corrected for flux partition between fluid and particles between the wall and the first flow grid point (Raupach, 1991; Shao and Li, 1999). The LBC for scalars (temperature and specific humidity) are flux-free and thus the only source/sink of heat and water vapor in the simulations is through the interaction of the flow with the saltating particles.

All simulations are performed on a grid of 64 x 64 x 128 grid points with a uniform grid in the horizontal directions and a stretched grid in the vertical. A stationary turbulent flow is allowed to first develop, following which, the snow surface is allowed to be eroded by the air. All physical constants and parameters along with additional details of the numerical setup are

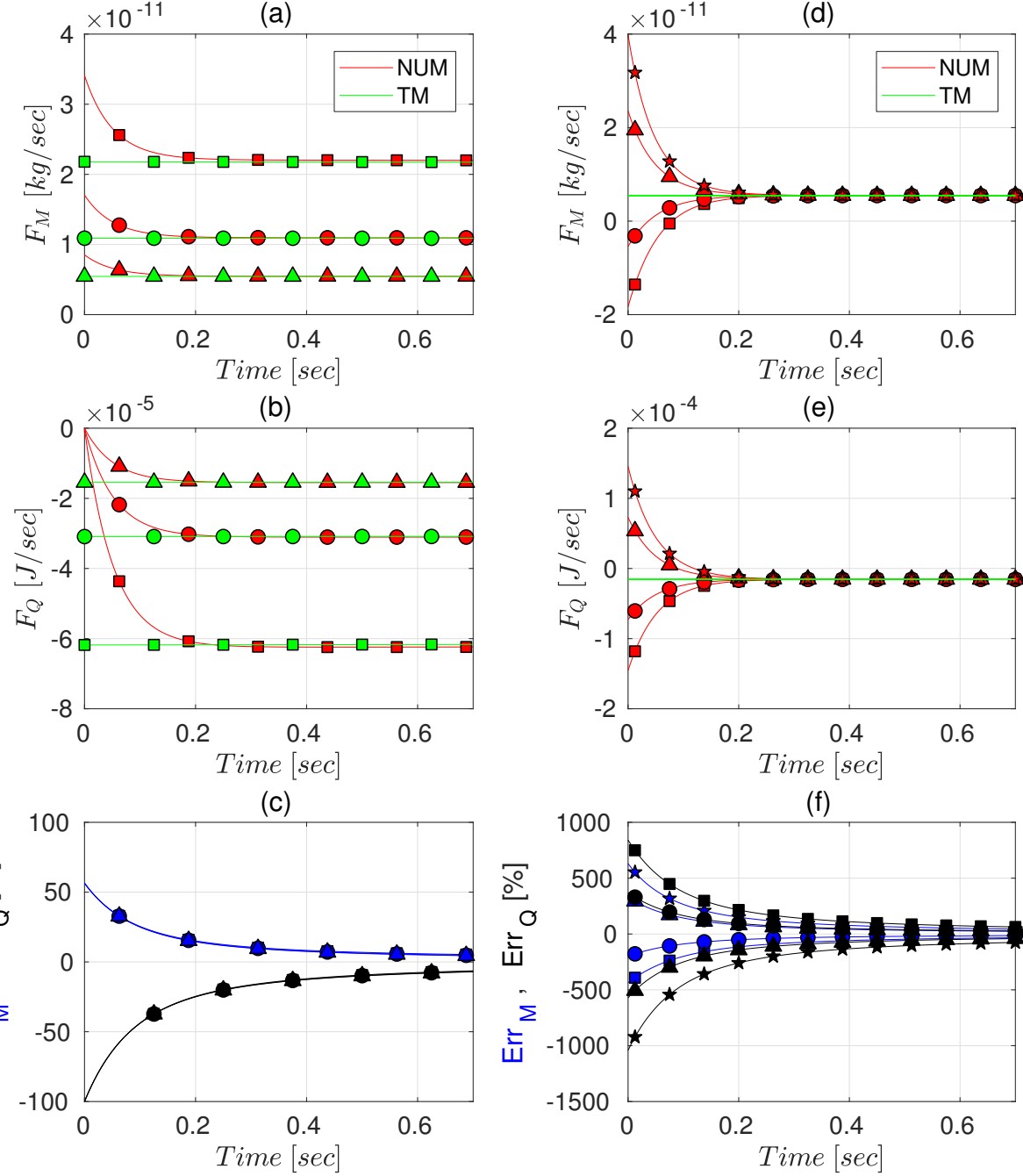

**Figure 1.** TM and NUM solutions for a particle of 200 $\mu m$ diameter in different environmental conditions. **Experiment I-A**: (a) Rate of mass and (b) heat output with (c) corresponding errors; $T_{p,IC}-T_{a,\infty} = 0$, $\sigma_* = 0.8$ (squares), 0.9 (circles), 0.95 (triangles). **Experiment I-B**: (d-f) same as (a-c) with $\sigma_*$=0.95; $T_{p,IC}-T_{Air}$= -2 K (squares), -1 K (circles), 1 K (triangles), 2 K (stars).

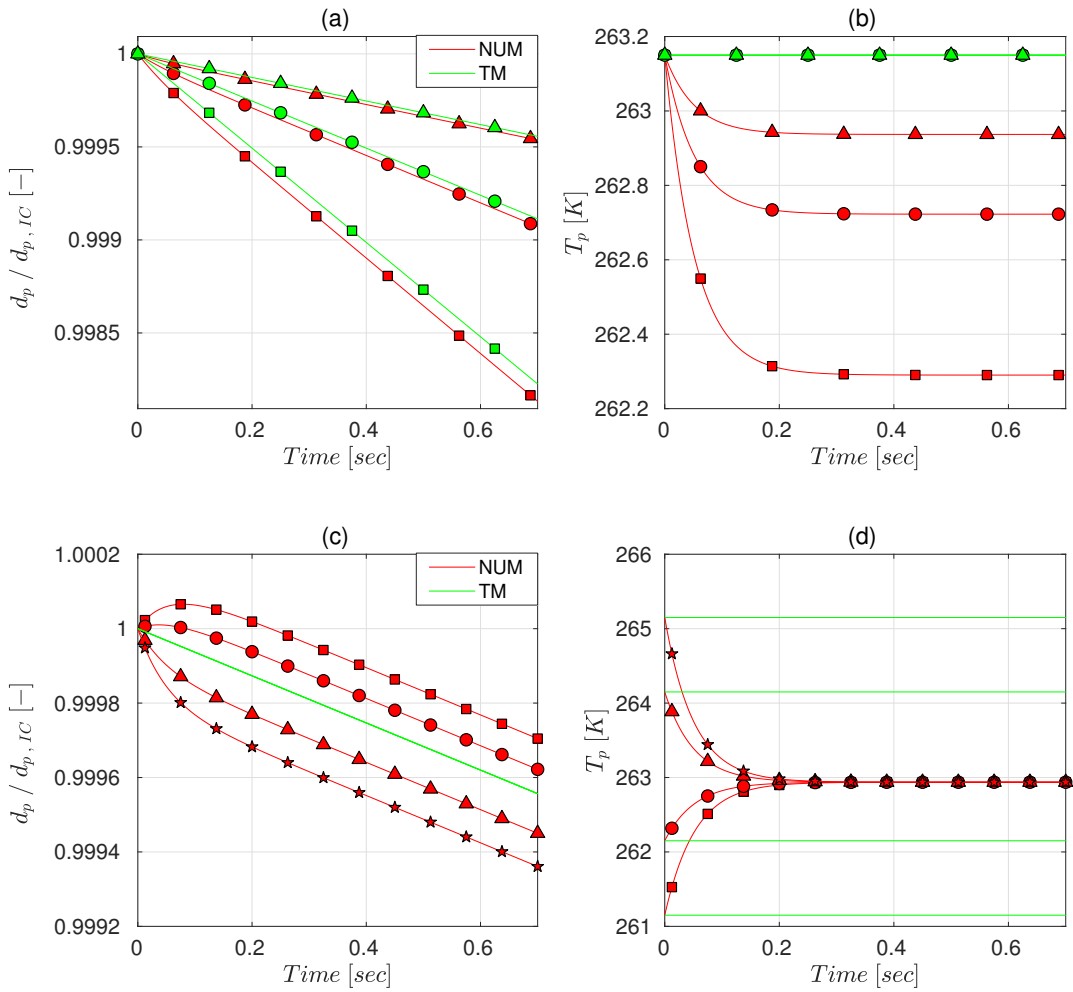

**Figure 2.** TM and NUM solutions for a particle of 200 $\mu m$ diameter in different environmental conditions. **Experiment I-A**: Evolution of particle (a) diameter and (b) temperature; $T_{p,IC}{-}T_{Air}{=}$ 0, $\sigma_*{=}$ 0.8 (squares), 0.9 (circles), 0.95 (triangles). **Experiment I-B**: (c-d) same as (a-b) with $\sigma_*{=}$0.95; $T_{p,IC}{-}T_{Air}{=}$ -2 K (squares), -1 K (circles), 1 K (triangles), 2 K (stars). Note that the particle diameters are normalized by the initial diameter of the particle ($d_{p,IC}$).

provided in Supplementary Material S2. The LES in the configuration used in this study resembles the classic case of LES of a channel flow common in computational fluid dynamics research. The LES code has been developed in-house for last many years beginning with (Albertson and Parlange, 1999) and has been used and validated for various atmospheric boundary layer problems such as flows over heterogeneous surfaces Bou-Zeid et al. (2004), hills (Diebold et al., 2013), diurnal cycles (Kumar et al., 2006), urban canopies (Giometto et al., 2016) and wind farms (Calaf et al., 2010; Sharma et al., 2017). The same code was used previously for modeling snow saltation by Groot Zwaaftink et al. (2014).

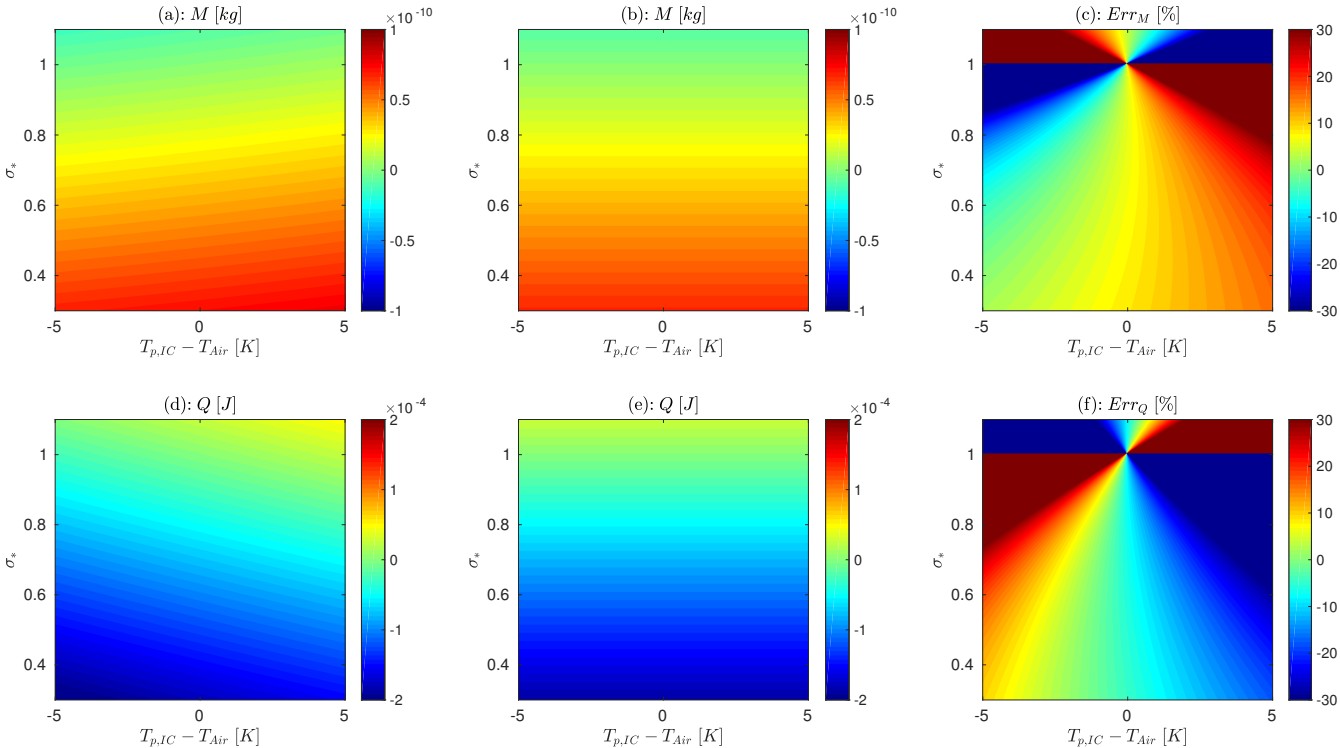

**Figure 3.** TM and NUM solutions for a particle of 200 $\mu m$ diameter in different environmental conditions. **Experiment II**: Total mass output during 0.5 seconds by the (a) NUM and (b) TM solutions with (c) corresponding error for $\{0.3 \leqslant \sigma_* \leqslant 1.1, -5\,K \leqslant T_p - T_{Air} \leqslant 5\,K\}$. Similar plots for total heat output presented in (d-f).

For the TM approach, Eq. (4) is used to compute the specific humidity and (by multiplying with the latent heat of sublimation) heat forcing due to each ice grain in the flow. On the other hand, for the NUM approach, Eqs. (1) and (2) are solved and only the turbulent transfer of heat between the air and the particle (second term in R.H.S of Eq. (1)) acts as a heat forcing on the flow. An implication of the NUM approach is that the particle temperature evolves during the ice-grain's motion and this necessitates providing an initial condition for the particle temperature ($T_{p,IC}$).

The principle aims of Experiment III are to firstly quantify particle residence times (PRT) and their dependence on wind speeds and relative humidities and secondly, compute the differences in the heat and mass output between the NUM and the TM approaches during saltation of snow with complete feedback between the air and the particles. PRT is defined as the total time the particle is air-borne and in motion, including multiple hops across the surface. Note that the PRT is not computed for particles in suspension, i.e, particles that stay aloft and never return to the surface. Towards this goal, simulations are performed, each with a combination of initial surface stress, $u_* \in \{0.4, 0.6, 0.8\}$ ms$^{-1}$ and initial saturation-rate, $\sigma_* \in \{0.3, 0.6, 0.9\}$. These values are classified as low (L), medium (M) and high (H) and correspond to wind speed at 1 m height above the surface of 11 m/s, 16.3 m/s and 21.8 m/s respectively. Note that during fully developed snow transport, the particles in the air

impart drag on the flow causing the flow to decelerate. The wind speeds at 1 m during fully developed saltation are 8.77 m/s, 11.34 m/s and 13 m/s respectively. The simulations are named as U$\alpha$-R$\beta$, where $(\alpha, \beta) \in \{L, M, H\}$. Each combination is simulated independently for the NUM and TM approaches resulting in a total of eighteen simulations. Experiment III is limited to simulating the *usual* case where the initial air temperature ($T_{Air,IC}$) is the same as $T_{p,IC}$.

Experiment IV is aimed at exploring the implications of differences between the two approaches in cases where $T_{Air,IC}$ is significantly different from $T_{p,IC}$. Such conditions can occur in nature during events such as marine-air intrusions, katabatic winds, spring-season saltation events and winter flows over sea-ice floes, where significant temperature differences between the air and snow-surface are likely. We repeat the low wind case of Experiment III with $u_* = 0.4 \mathrm{ms}^{-1}$ and choose the initial saturation-rate to be 0.95, motivated by results in Experiment II where errors were found to increase with increasing saturation-

rate. Simulations (named as UL-T($\gamma$), where $T_{Air,IC} - T_{p,IC} = \gamma$) are performed once again for each of two approaches with $\gamma \in \{\pm 1\,\mathrm{K}, \pm 2.5\,\mathrm{K}, \pm 5\,\mathrm{K}\}$ resulting in a total of twelve simulations. In all simulations performed for Experiments III and IV, $T_{p,IC} = 263.15\,\mathrm{K}$. It is important to note that the initial condition for particle temperature ($T_{p,IC}$) is fixed throughout the simulation period, which essentially means that surface temperature is kept constant. This is consistent with the imposed zero flux of heat at the surface. This imposition will be justified *a posteriori* in the following section.

## 3.2   Results

In this section, results from the LESs performed for Experiments III and IV are presented. Note that only the relevant results are presented, namely (a) particle residence times as a function of particle diameters and different forcing setups and (b) differences between the NUM and TM approaches for calculating average mass and heat transfer rates during saltation. Other results, for example, vertical profiles of mean and turbulent quantities, although interesting are relegated to the supplementary material as

their analysis is out of scope of the current work. Additionally, two video illustrations (Supplementary Movie M1 and M2, see Supplementary Material S4) of an LES is provided to help visualize and frame the context of the simulations performed.

### 3.2.1   Particle Residence Times versus $\tau_{relaxation}$

As mentioned in the concluding lines of the Sect. 2, the principle quantity of interest is the PRT of saltating ice grains. In Fig. 4a, the mean and median PRT of five different simulations of Experiment III are shown as a function of the particle diameter.

Additionally, values of $\tau_{relaxation}$ computed in Experiment I for wind speeds ranging from 0 to 10 ms$^{-1}$ are also shown in the shaded region. Recall that the shaded region represents all the plausible values of $\tau_{relaxation}$ in naturally-occurring aeolian transport. As examples, $\tau_{relaxation}$ trends for 3 wind speeds, 0, 1 and 10 ms$^{-1}$ are shown and the power-law dependence can clearly be seen. It is found that $\tau_{relaxation}$ is comparable to the PRT of saltating grains with diameters between 125 and 225 $\mu$m. For 200 $\mu$m, the mean PRT is found to be 0.6 seconds while the median PRT is 0.2 seconds, which is outside the range of

admissible values of $\tau_{relaxation}$. For particles larger that 225 $\mu$m, the PRTs are an order of magnitude smaller than plausible values of $\tau_{relaxation}$ and therefore the TM model is likely to provide wrong values of mass loss. On the other hand, lighter particles with diameters smaller than 100 $\mu$m have much longer PRTs and the TM model is therefore valid. This proves that while the TM model is applicable for a majority of particles in suspension, it is likely to cause errors for particles in saltation.

Results presented in Fig. 4a provides two additional insights. Firstly, it is quite interesting to note that particles larger than 100 $\mu$m have the same mean PRT irrespective of low, medium or high mean wind speeds. This means that the dynamics of the heavier particles are unaffected by different mean wind speeds simulated in Experiment III, which is consistent with the notion of self-organized saltation, which has recently been shown by Paterna et al. (2016). For particles smaller than 100 $\mu$m, the mean PRTs increases with wind speed. Secondly, the initial saturation-rate does not seem to effect the PRT statistics for medium and high wind conditions and the UM- and UH- curves for different R values overlap ( this is the reason only five PRTs are shown in Fig. 4a). In these cases turbulence is sufficient to rapidly mix any temperature anomaly due to sublimation throughout the surface layer. On the other hand, the mean PRTs of particles smaller than 75 $\mu$m, decrease with decreasing initial $\sigma_*$. A preliminary hypothesis is that in low wind conditions (UL), low initial saturation-rate results in more sublimation and cooling near the surface, resulting in suppression of vertical motions. Even though this is an interesting result, further research is needed to definitively link drifting snow sublimation to lower PRTs. We leave further exploration of this phenomenon for a future study with some preliminary analysis provided in Supplementary Material S3.

The PRT distributions are found to be quasi-exponential with long tails, thus resulting in large differences in mean and median PRTs shown in Fig. 4a. These distributions are also strongly dependent on the particle diameter. As an illustration, in Fig. 4b, cumulative distributions of PRTs are shown for four particle diameters along with the corresponding range of plausible $\tau_{relaxation}$ values. For the mean particle diameter of 200 $\mu$m, we find that between 65% to 85% of particles have PRTs shorter than $\tau_{relaxation}$, whereas for the 75 $\mu$m particles, at most 30% particles lie below the maximum $\tau_{relaxation}$ threshold. This reinforces the fact that applying the steady-state TM solution to sublimating ice-grains in saltation could be potentially erroneous.

### 3.2.2 Differences in total mass loss between NUM and TM models

We can directly assess the implications of differences in grain-scale sublimation between the two approaches on total mass loss rates during saltation at larger spatial scales as simulated using LES in Experiments III and IV. In Fig. 5, we compare the total 15-min averaged rate of mass loss computed in all cases in Experiment III (subfigure a) and Experiment IV (subfigure c) using the NUM and the TM approaches with corresponding errors shown in subfigures b and d respectively. Recalling the adopted convention of +(-) as gain(loss) of flow quantities, it can be seen in Experiment III, that sublimation increases with $u_*$ and decreases with $\sigma_*$. The errors on the other hand increase with increasing values of both $u_*$ and $\sigma*$. The increase in error with $u_*$ is mostly due to the fact that an increase in $u_*$ proportionally increases the total mass entrained by air (see Supplementary Fig. S1). The increase in error with increasing $\sigma_*$ is in accordance with analysis done in Experiment II ( see Fig. 3(c,f) ) where it was shown that the NUM and TM solutions diverge with increasing saturation-rate. The least error, 26% is found for case UL-RL (i.e., $u_* = 0.4$, $\sigma_* = 0.3$) while the largest error, 38% is found for UH-RH ($u_* = 0.8$, $\sigma_* = 0.9$). Overall, for all the simulation combinations, the NUM approach computes larger mass-loss than the TM approach.

Experiment IV highlights the effect of temperature difference between particle and air on sublimation. As shown in Fig. 5c, the mass output is found to be negative (deposition) for the NUM solutions when the air is warmer than the particles (i.e., cases UL-T($\gamma$) with $\gamma > 0$ ). This is contrary to the TM solutions which indicate sublimation. In cases with $\gamma < 0$, the NUM

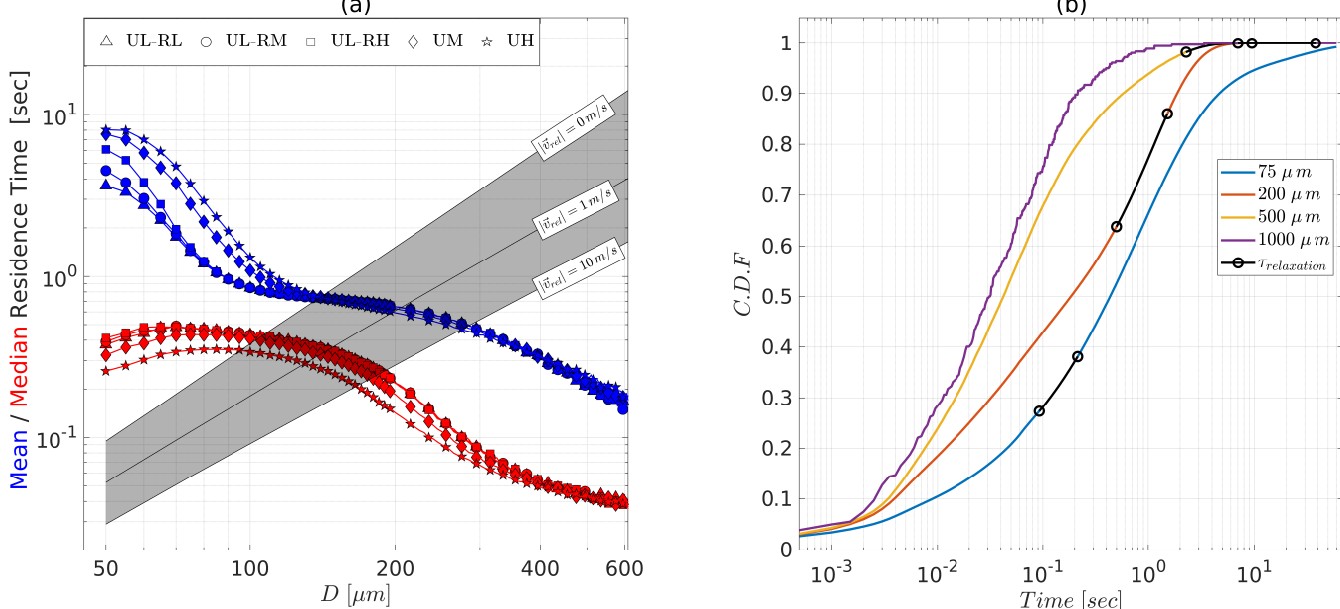

**Figure 4.** (a) Mean and median particle residence time (PRT) as a function of particle diameter. The plausible values of $\tau_{relaxation}$ are represented by the shaded region with trends for three values of $|\boldsymbol{u}_{rel}|$ shown by straight lines. Note that the horizontal axis is logarithmic. (b) Cumulative Distribution Functions of PRTs for four particle diameters along with range of plausible $\tau_{relaxation}$ values marked by overlying black curves

approach shows a much higher sublimation rate than the TM solutions. This occurs firstly due to higher vapor pressure at the grain surface that results in enhanced vapor transport and secondly because the warmer particles heat the surrounding air via sensible heat exchange, causing the relative humidity to decrease. Errors increase dramatically from an already high 100% for UL-T(+1) to 800% for UL-T(+5). Simulations performed for medium and high wind cases in Experiment IV showed even
5    higher errors, similar to results in Experiment III and are shown here.

## 4    Discussion and Conclusion

In this article, we revisit the Thorpe and Mason (1966) model used to calculate sublimation of drifting and blowing snow and check its validity for saltating ice grains. We highlight the fact that solutions to unsteady heat and mass transfer equations (NUM solutions) converge to the steady-state TM model solutions after a *relaxation* time, denoted as $\tau_{relaxation}$ that has a power-law
10   dependence on the particle diameter and is inversely proportional to the relative wind speed. Through extensive LESs of snow saltation, we compute the statistics of the PRTs as a function of their diameters and find them to be comparable to $\tau_{relaxation}$. This helps explain the difference between mass output when using the NUM model to the TM approach, also computed during the same LESs. The NUM approach computes higher sublimation losses ranging from 26% in low-wind, low saturation-rate

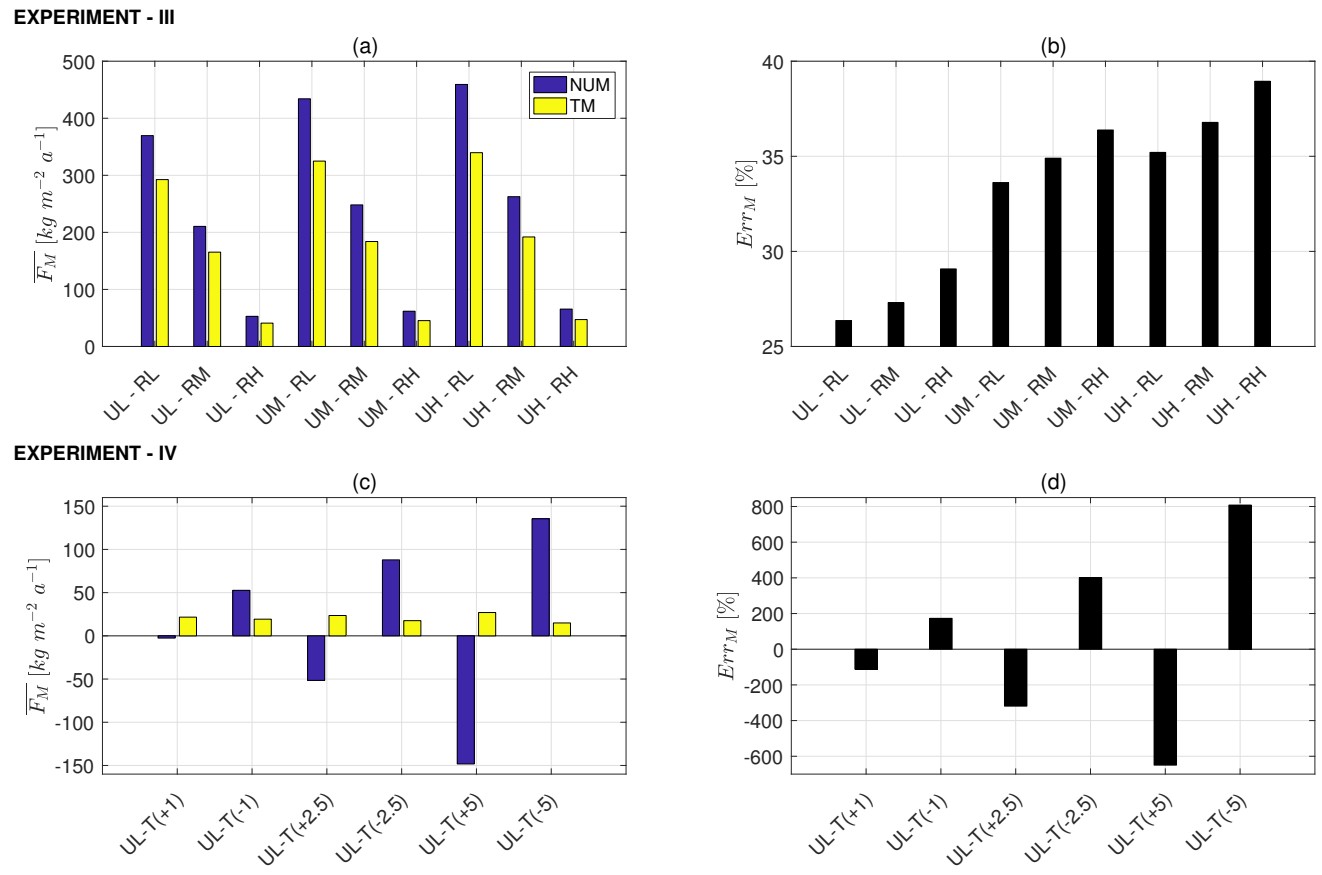

**Figure 5.** Experiment III: (a) Average rate of mass loss during 15 minutes of saltation, (b) Error between NUM and TM solutions. Corresponding plots for Experiment IV in (c) and (d) respectively. Note that the units used for rate of mass loss are kilograms per unit area per unit year.

conditions to 38% in high-wind, high saturation-rate conditions. Another set of numerical experiments explore the role of temperature differences between particle and air temperature in inducing differences between NUM and TM solutions. We find the effect to be extremely dramatic with errors of 100% for a temperature difference of 1 K with increasing errors for larger temperature perturbations. In general, the two solutions are found to diverge rapidly as the saturation-rate tends towards 1. The
5   results showing differences of mass output between the NUM and TM approaches in the LESs in Experiments III and IV, with complete feedback between particles and the air are thus shown to be closely correlated to the results from extremely idealized simulations of heat and mass transfer from a solitary ice grain in Experiments I and II.

     The LES results do come with a few important caveats. Firstly, the temperature and specific humidity fluxes at the surface are neglected. In other words, particles lying on the surface are considered to be dormant and do not exchange heat or mass with the
10   air. A corollary to neglecting the scalar fluxes at the surface is that the initial condition for temperature of the particles entering

the flow is fixed. This may be justified by considering that during drifting and blowing snow events, the friction velocity at the surface drops dramatically. This fact has been observed in both in experiments (Walter et al., 2014) and in our current LESs (see Supplementary Fig. S2). This implies that direct turbulent exchange between the surface and air is curtailed and instead, the dominant exchange occurs between air-borne particles and the air. In fully-developed snow transport events, this is most likely

to be true and only in intermittent snow-transport events will the surface fluxes be relatively important. This is nevertheless an important assertion that shall be more closely examined in future studies involving a full surface energy balance model, where the evolving temperature of the saltating ice grains, prior to deposition is taken into account while calculating snow-surface temperatures.

Further work is required to make concrete improvements to modeling of sublimation of saltating snow, especially in large-

scale models that do not explicitly resolve saltation dynamics. One potential approach is to modify the Monin-Obukhov based lower boundary conditions for heat and moisture to account for particle temperature during blowing snow events. An ancillary outcome of this study is the discovery that buoyancy can affect the dynamics of lighter snow particles (with diameters less than 75 $\mu$m) and decrease their residence times. Investigating this phenomenon requires a detailed analysis of turbulent structure within the saltation layer and is left for future publications.

In conclusion, analogous to the role played by saltating grains in efficient momentum transfer to the underlying granular bed, the NUM approach can be considered as an efficient transfer of heat and mass between the flow and the underlying snow surface, albeit with a closer physical relationship between the thermodynamics of the snow surface and that of the air. Thus, along with momentum balance of blowing snow particles, particle temperature and its thermal balance must also be taken into account.

*Acknowledgements.* We acknowledge the support of the Swiss National Science Foundation (Grant n. 200021-150146) and the Swiss Super Computing Center (CSCS) for providing computational resources (Project: s633). We additionally thank Marco Giometto for providing the original version of the LES code and Hendrik Huwald, Tristan Brauchli, Annelen Kahl and Celine Labouesse for illuminating discussions and important suggestions in improving the manuscript. No new data were used in producing this manuscript.

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
