# Peer review of "On the suitability of the Thorpe-Mason model for Calculating Sublimation of Saltating Snow"

_The Cryosphere, 2018_

## Referee Comment (RC1) · Anonymous Referee #1 · 10 Apr 2018

General Comments:

This is a very interesting and valuable paper that pursues an important topic in blowing snow studies. The manuscript addresses the suitability of the steady-state sublimation model of Thorpe and Mason using a high degree of modeling complexity. The manuscript is also unique in that it addresses the issue of snow sublimation at a much higher resolution than the typical hydrological modeling style. This work should be published.

However, the increased sophistication of the modelling techniques requires further clarification because of unanswered questions of how certain sub-processes of saltating particle sublimation are handled numerically. By nature of this paper's numerical approximation of reality, it is the onus of the authors to clearly explain how their models

(and their required assumptions) differ from the old model (and its assumptions). Given that none of this work is grounded in observation, a higher level of transparency is required to make this work truly beneficial.

The transient nature of these processes at this time scale are the most important result of this manuscript and require significant explanation of what is actually occurring in the model. One significant of LES is the ability to represent some sort of turbulent structure in an effort to more closely represent reality. As such a more thorough explanation of the turbulence driving the NUM model is needed to be of utility to researchers attempting to apply these findings in a real-world setting. This is especially true because of the parameterization of Nusselt and Sherwood numbers used to drive sublimation rates.

The experiments describe using a variety of particle sizes but does little to mention how this affects of modes of transport (and sublimation according to "mode"). Particle flux profiles and sublimation profiles with height would further inform the realism of this study. This could be put in supplementary material without serious restructuring of the manuscript.

Specific Comments:

P1 L8 (and throughout): This does not appear to be a perturbation in a functional sense as you are not perturbing a system. This is more like a sensitivity analysis, changing initial conditions. There is ambiguity in this phrasing as a perturbation of 1K can mean strictly a temperature difference of 1K (which I believe you mean) or adding 1K to the difference. I would suggest replacing perturbation so as to not mislead the reader into thinking they will be reading a manuscript using perturbation theory.

P2 L1: There are actually three modes of transport, including creep.

P3 L20: Is this truly a representative illustration? What is the ventilation rate of that bottle?

P4 L1: That's very true!

\*\* P4 L15-17: This is a very important change of sign. What is the explanation for the initial change from deposition to sublimation for the colder-than-air particles? Was this a period where the particle actually warmed up? Gained mass? Was the air surrounding the particle cooling through latent or sensible heat? It is very exciting that this information is finally available!

\*\* This concept is overlooked throughout the paper. You have a wonderfully extensive data set. Explain whether or not sublimation is the only transfer of energy to your saltating particles. Please explain whether or not (and why) the particles actually warm in your simulations. Is there no thermodynamic feedback on the systems in section 2? Does Sigma star never change with time? Why or why not? Do values in Fig 1b,e actually affect the change in the ambient or near-particle air, or did the model assume these energy exchanges were "advected" away?

P4 L21-24: This is an interesting idea. However, there appears to be some serious assumptions used to reach this conclusion. Please clarify the following: Did you assume there is no ventilation or sublimation of particles when they are in contact with the surface, and is it assumed that the wind speed is constant across all heights of the trajectory of a saltating particle? Admittedly, there is certainly a connection between relaxation times and residence times, but it would increase the quality of the paper to convey either what assumptions are necessary to make the conclusions from Figure 2a to be truthful?

P4 L21: Relaxation time is great, but what about the time that $Err_{Q,M}$ goes to zero (Fig 1f)? This value seems equally as important, as it appears to be a lower bound on the timeframe in which stationary wind/transport conditions are required to allow all the numerical errors to cancel out. This paper would benefit from an exploration (surface plot) of relaxation time over the parameters ($\sigma_*$, $T_P - T_{Air}$), and supplement Fig 1 a-c very well.

P4 L27: Can you speculate as to what is the physical (or numerical) meaning of this

scaling relationship? Or is this a purely empirical finding?

P4 L31 Fig 1g-l: Please expand the negative range of $T_P - T_{Air}$. There are environments where föhn events can bring dramatic changes of temperature up to 10C over only a few hours!

P5 L20: Look at parameterization

P5 L26: It is unclear to me how using this stationary flow is fundamentally different from your steady state model. Was the LES used because it is a more sophisticated framework in which to calculate these fluxes? Besides the evolution of friction velocity is Figure S2, I am afraid I have missed the point of using such a complex tool to solve some PDE's.

P5 L32: Why was this not discussed in the previous experiments?

P6 L3: Please stop calling this realistic saltation of snow. The parameterizations and assumptions necessary to run this numerical model make this statement misleading. Please rephrase as LES simulation of saltation or something similar.

P6 L5: These varying friction velocities are referred to as "low medium and high wind-speeds" in line 33. What wind speeds were necessary for these values? Friction velocities do a poor job of representing turbulence in even subtly complex terrain, and as saltation is a drag-driven process, at the very least, mean wind speeds should be included in the manuscript, and extensive time series of turbulence statistics (Turbulence Intensity, TKE, shear stress, etc.) in the supplementary material. As this research is conducted to benefit those that models in natural settings, and those natural setting will be much more turbulent than the LES, and that turbulence is what is driving the ventilation rates, more information about the model is needed.

P6 L5: Where is "the surface" defined for this stress calculation? And how is that surface defined? How can that be reconciled with the fact there is windpumping into the snow pack? Or is this a Reynolds-stress-based value?

P6 L15: Why a different range of temperatures than Section 2?

Fig 2a: Redo the plots so it is clear what is happening. I cannot understand anything from 200um to 1000um. The diameter plot markers appear somewhat logarithmically. Try plotting with a logx scale? And why do the residence time measurements become more sparse at smaller particle sizes? Please redo the symbols as they are confusing and inconsistent, or eliminate them altogether.

P6 L23: There is no dependence on $\sigma_*$?

P6 L29: Please elaborate why the values of mass loss are wrong. It appears in Fig 1c,f that the cumulative errors go to zero over time. Why is this no longer the case with LES?

P7 L1: Please rephrase "larger-scale turbulence statistics." It unclear to me how any "larger-scale turbulence" can be represented in a 6x6x6 meter box. Is this not an increase in mean windspeed?

P7 L3-8: This is a very interesting finding! This suppression of vertical motions and how the model responds should be elaborated on! A comparison of the vertical turbulence statistics amongst the experiments is necessary as they all assume uniform initial air temperature (P6 L5 comment). How does vertical mixing in the LES deal with this over time? Logic would imply that this same suppression of vertical mixing could also be caused by a colder snow surface temperature and increased stability. Why have you disregarded particle surface temperature in your PRT experiments? Would this effect be found if a temperature gradient as found in nature were present, or would the numerical effect be overwhelmed by the near-surface temperature gradients? As it stands, this statement cannot stand alone and the conclusion needs more development and supporting data/plots would be very beneficial.

P7 L3-8 These are very small particles, can they be considered in "Suspension?" Obviously, there is a full spectrum of motions, but approximately where have other researches been separating saltation from suspension on Fig 2a? This would be very informative as the paper by nature is a saltation study.

P7 L12: Very exciting finding!

P7 L18: What is field scale?

P7 L21: Can anything be said about the low end of the friction velocity domain where intermittent transport dominates? Would TM over or underestimate in that case?

Fig 3: Where have the particle diameters gone? What distribution of sizes are you using?

P8 L22: Not perturbations.

Overall this is will be a great contribution to the field, and will no doubt be referenced extensively. However, at its current state, the manuscript needs to be expanded. It is much too short for the gravity of the conclusions.

Technical Corrections:

P1 L7: Please specify: snowpack surface temperature, snow particle surface temperature?

EQ3 What is "d"? $d_p$?

P2 L24 "Saturation $\sigma_*$..." ? Do you mean sigma is saturation?

P2 L27: Add space after sentence end.

P5 L12: "an erodible"

---

## Referee Comment (RC2) · Anonymous Referee #2 · 28 May 2018

General comments:

The manuscript by Sharma et al. presents a novel method to account for the effects of sublimation on blowing snow in the atmospheric boundary layer, based on recent high-quality Large-Eddy Simulation (LES). Of particular focus is the sublimation processes in the saltation layer; although numerous attempts have been done to reveal the sublimation processes in the suspension layer of blowing snow, investigations focused on the saltation layer were scarce. They found that regarding the saltation layer, the Thorpe and Mason model (a general method used in blowing snow sublimation) could yields erroneous results. Their findings will be of great importance to clarify an appropriate application condition and limitation of the Thorpe and Mason model. Additionally, the newly proposed method in the manuscript may become important in the

developing a better understanding of blowing snow sublimation.

The manuscript overall is very well written. The research is of good quality, LES runs with a Lagrangian blowing snow model are impressing. The methodologies and discussions are organized properly in general. Supplementary material is quite informative. Thus in general I think this work is highly appropriate for publication in The Cryosphere (TC). Additionally, the subject fits very well the journal.

I think it would be useful if the authors gave more information about the vertical profiles, in the latter calculations (EX. III and IV). I am wondering how the profiles such as wind speed, mass flux, temperature, sublimation rate etc. will change during saltation (e.g. before saltation, after the onset (transient regime), during steady state). I understand that investigation of the vertical structures is not the primary purpose of this study, however some interpretation of the vertical profiles seems essential. This will increase the credibility of the model.

Specific comments:

P.2, L.5: Perhaps it will be a good idea to refer the previous sublimation simulations in suspension. e.g., Xiao et al., 2000, An intercomparison among four models of blowing snow, Boundary-Layer Meteorology, 97, 109-135.

P2., L.9: "recent studies using high-resolution large-eddy simulations" – is the reference really use LES? I could not confirm in that paper.

P.2, L.24: "saturation ${\sigma_*}$" – does it mean rate of saturation?

P.2, L.25: "Groot Zwaaftink et al. 2011, 2014" should be "Groot Zwaaftink et al. 2011". In Groot Zwaaftink et al. (2014), mass loss due to sublimation are neglected in the calculation.

P.2, L.27: "Vionnet et al., 2014).In" should be "Vionnet et al., 2014). In" (please add a space).

P.3, L.9: The units should be given in Roman type, I think.

P.4, L.3: Is the first-order scheme sufficient in the calculation?

P.5, L.12: " a erodible" should be "an erodible".

P.6, L.15: Do you have any specific reason for the different temperature conditions (-5<dT<5 in EX. IV, -2.5<dT<2.5 in EX. II).

P.7, L.7-8.: "low initial saturation results in more sublimation and cooling near the surface, resulting in suppression of vertical motions." – This is interesting indeed. Could you show the modification of the vertical profiles (temperature, sublimation rate, wind speed etc.) to illustrate these processes?

P.7, L.18: "at a field scale" specifically, during realistic saltation of snow?

P.8, L.2: "here)." should be "here." (an unnecessary parenthesis).

(Supplementary material)

P.2, L.26 (S9): "ln" should be given in Roman type.

P.3, L.27: I think parenthesis is missing around the reference.

P.3, L.29: {\sigma_d_p} is undefined, I think.

P.4, L.2-4: Could you include the relevant references (Clifton and Lehning (2008) ?).

P.4, L.16-17: Is the rebound angle the same as that for sand? I think Kok and Renno (2009) obtained the results for sand. Do you hypothesis the angle is similar to sand saltation?

P.4, L.19-20: "dislodge additional additional particles" should be "dislodge additional particles".

P.5, L.22: "represented by f." f should be given in italic.

P.7, L.23: Is the first-order scheme adequate for the computation in this study?

P.8. "Time step" – All the elements (fluid, particle, and scalars) have the same timestep?

P.9, L.5: "It shows that that once" should be "It shows that once"?

---

## Author Comment (AC1) · 31 Aug 2018

**Response to reviewers:**
**On the suitability of the Thorpe-Mason model for calculating sublimation of saltating snow**

Varun Sharma, Francesco Comola and Michael Lehning

August 31, 2018

**A note to all reviewers**

Please see the additional document found in the "comments_to_all.pdf" file

**Response to Reviewer # 1**

**Opening Remarks:**
We would like to thank Reviewer #1 for his/her detailed critique of the submitted manuscript and for asking different clarifications and questions. Broadly stating, the following additions and/or corrections were made to the article in response to Reviewer #1's comments:

- We has enlarged Section 2 with a more detailed description of the dynamics of the heat and mass transfer from a solitary ice-grain and made clear the approximations entailed.

- An new figure is added that shows the evolution of particle diameter and temperature is Experiment I.

- A visual representation of one of the LES performed in Experiment III is provided to make clear, the sort of LES that have been performed.

- The supplementary material has been updated with 5 additional figures detailing various results from the LES.

- The caveats and limitations of the current LES model setup have been more explicitly mentioned in the updated manuscript and a few future directions of research have been listed in the expanded concluding section of the manuscript.
* * *
**A: Scientific Concerns**

- **A.1 : P1 L8 (and throughout): This does not appear to be a perturbation in a functional sense as you are not perturbing a system. This is more like a sensitivity analysis, changing initial conditions. There is ambiguity in this phrasing as a perturbation of 1 K can mean strictly a temperature difference of 1 K (which I believe you mean) or adding 1 K to the difference. I would suggest replacing perturbation so as to not mislead the reader into thinking**

*they will be reading a manuscript using perturbation theory.*

**Response A.1:** We thank the reviewer for this comment and agree that perhaps using the word "perturbation" is misleading. We modify the text as follows:

*With a small temperature difference of 1 K between the air and the snow surface, the errors due to the TM model are already as high as 100% with errors increasing for larger temperature differences.*

————————————————

- **A.2 : P2 L1: There are actually three modes of transport, including creep.**

  **Response A.2:** We agree with the comment and the text has been modified as follows:

  *Aeolian transport of snow can be classified into three modes, namely, creeping, saltation and suspension. Creeping consists of heavy particles rolling and sliding along the surface of the snowpack either due to form drag or bombardment due to impacting particles.*

  ————————————————

- **A.3 : P3 L20: Is this truly a representative illustration? What is the ventilation rate of that bottle?**

  **Response A.3:** This analogy was used only to highlight the fact that there is a possibility of deposition of vapor on saltating ice grains. This possibility has never been explored and/or accounted for in the existing models that only assume a uni-directional exchange of water mass from the ice grain to the atmosphere (unless there is super-saturation of the atmosphere, which is usually not allowed in atmospheric models). On a beer bottle or anything from a refrigerator, we see a reverse process of extraction of vapor from the atmosphere onto the material and the atmosphere does not need to be super-saturated for this !

  In terms of ventilation rate, if we consider the Reynolds' number of a beer bottle of diameter 5 centimeters with a ice grain of diameter 200 microns, there is approximately two order of magnitude of difference. To have the same Reynolds number, $|\vec{u}_{rel}|^{icegrain} \approx 250|\vec{u}_{rel}|^{beer\ bottle}$. Thus, for a typical relative velocity between a saltating ice grain and air of 5 m/s, the beer bottle's relative velocity needs to be only 0.02 m/s for the same Reynolds number and exchange coefficients. This is entirely plausible. Thus, even in terms of Reynolds numbers, the beer bottle analogy works !

  ————————————————

- **A.4 : P4 L1: Thats very true!**

  **Response A.4:** *Skipped*

  ————————————————

- **A.5 : P4 L15-17: This is a very important change of sign. What is the explanation for the initial change from deposition to sublimation for the colder-than-air particles? Was this a period where the particle actually warmed up? Gained mass? Was the air surrounding the particle cooling through latent or sensible heat? It is very exciting that this information is finally available! This concept is overlooked throughout the paper. You have a wonderfully extensive data set. Explain whether or not sublimation is the only transfer of energy to your saltating particles. Please explain whether or not (and why) the particles actually warm in your simulations. Is there no thermodynamic feedback on the systems in section 2? Does Sigma star never change with time? Why or why not? Do values in Fig 1b,e actually affect the change in the ambient or near-particle air, or did the model assume these energy exchanges were "advected" away?**

**Response A.5:** We thank the reviewer for posing several critical points in this question. These questions strike at the heart of the message of the paper and thus it is extremely important for us to make sure that we are able to get our message across to the readers.

In the first set of experiments in Experiment I, the particle as well as the air have the same temperature of 263.15 K. However, the air is not saturated and thus there is a diffusion of mass from the ice grain to the air as described by Equation 2. However, since the temperature of the ice grain is the same as the air, there is no heat transfer. The initial energy for the sublimation must then come from the internal energy of the ice grain. The internal energy is nothing but the heat energy stored in the ice grain as represented by the grain temperature. As the internal energy of the ice grain is consumed, it's temperature decreases and as soon as this happens, heat transfer between the ice grain and the air commences. After a transient period, an equilibrium condition is achieved where the particle temperature becomes constant and all the energy necessary for sublimation comes directly from the atmosphere.

The Thorpe-Mason model neglects the initial consumption of internal energy for sublimation and instead assumes that all the energy for sublimation comes from the atmosphere. In fact, the Thorpe-Mason model, by means of further approximations, does not consider particle temperature at all ! In this manuscript we show that for ice-grains in saltation, it is important to take into account, the ice-grain temperature and its evolution.

Returning to Experiment I, in the second part, we vary the initial temperature of the ice-grain with the ice grain being warmer or colder than the surrounding air. Here, the interpretation become slightly more difficult. In the case where the particle is colder than the air, there is both the warming of the particle as well as deposition. The particle gains energy both from convective heat transfer ( second term in the RHS of Eq 1 ) as well as gains mass (Eq 2). At a certain point in time however, the particle becomes warm enough ( though still colder than air ), that it begins to sublimate.

Note that the temperature ($T_{Air}$) and saturation (represented by $\sigma_*$) of the air surrounding the ice-grain does not change and all mass or energy gain/loss of quantities in the air as assumed to be advected way. We justify this because we considering the dynamics of a solitary ice grain, subjected to relatively strong air motions. A helpful image is to imagine a special hair-dryer blowing air onto a 200 micron ice grain. However, in the LES experiments in Section 3, all the feedbacks are taken into account.

Thus our motivation for Section 2 was to simply highlight the fact that particle temperature, and the coupled heat and mass exchange dynamics are important to account for, instead of the approximate solution presented by the Thorpe-Mason approach, particularly for the

[Figure]

Figure 1: NUM and TM solutions for a particle of 200 $\mu m$ diameter in different environmental conditions. **Experiment I-A**: Evolution of particle (a) diameter and (b) temperature; $T_{p,IC}-T_{Air}= 0$, $\sigma_*= 0.8$ (squares), 0.9 (circles), 0.95 (triangles). **Experiment I-B**: (c-d) same as (a-b) with $\sigma_*$=0.95; $T_{p,IC}-T_{Air}$= -2 K (squares), -1 K (circles), 1 K (triangles), 2 K (stars). Note that the particle diameters are normalized by the initial diameter of the particle ($d_{p,IC}$).

short time-scales that we are interested in.

*In response to the points raised in A.5, we have decided to update Section 2 to be more explicit about the nature of the simulations performed and the simplifications of the experiments. We have split Figure 1 of the original manuscript into two independent figures (a figure each for Experiment I and II) so that the plots are more clear and add an additional figure (Figure 1 in this document) to describe the evolution of particle diameter and temperatures in the different experiments. The new figure is added below for reference. The change in the text can be seen in the updated manuscript with the differences highlighted.*
* * *
- **A.6 : P4 L21-24: This is an interesting idea. However, there appears to be some serious assumptions used to reach this conclusion. Please clarify the following: Did you assume there is no ventilation or sublimation of particles**

*when they are in contact with the surface, and is it assumed that the wind speed is constant across all heights of the trajectory of a saltating particle? Admittedly, there is certainly a connection between relaxation times and residence times, but it would increase the quality of the paper to convey either what assumptions are necessary to make the conclusions from Figure 2a to be truthful?*

**Response A.6:** We thank the reviewer for raising a pertinent point here and giving us a chance to clarify. Firstly, we do not make a conclusive statement as evidenced by the use of the word "likely". Since there is no actual data on particle temperatures measured in wind tunnels or in the field at present, we cannot make a conclusive statement and more research is needed. Thus it is only a conjecture at present. The results in Experiments I and II however are well-correlated to those from LES data in Experiment III and IV and thus there is credible support for this idea.

It is true that we do not take into account, the particle temperature and sublimation while it is at rest at the surface. The heat and mass transfer from the particle to the air begins only once it is lifted from the surface (either aerodynamically or due to splash entrainment). Secondly, it is indeed true that the wind speed is not constant across all heights of the trajectory of the saltating particle. This is the reason why we compute the relaxation time from relative velocities ranging from 0 to 10 m/s. These would correspond to the upper and lower bounds of the relaxation time for particle heat and mass transfer dynamics. We compare the mean and median residence times of the saltating particles to this "range" of relaxation times (the shaded region in Figure 2a in the original manuscript) rather than a single value of relaxation time.
* * *
- *A.7 : P4 L21: Relaxation time is great, but what about the time that $Err_{Q,M}$ goes to zero (Fig 1f)? This value seems equally as important, as it appears to be a lower bound on the timeframe in which stationary wind/transport conditions are required to allow all the numerical errors to cancel out. This paper would benefit from an exploration (surface plot) of relaxation time over the parameters ($\sigma_*$, $T_P - T_{Air}$), and supplement Fig 1 a-c very well.*

**Response A.7:** This is great observation by the reviewer. However, we would like to point out that the errors in the cumulative heat and mass output in Figure 1c and 1f go to zero "very slowly" and in fact does not go to zero within typical saltation residence times. The quantity of relaxation time as we have defined is a far more robust measure to identify from simulations. It is also a more conservative measure as any particle with residence time lower than the relaxation time will, by definition, be lower than the measure proposed by the reviewer.

As far as the exploration of the relaxation time over the parameters goes, we did in fact do this exploration. However ,it was found that the relaxation time depends only on the particle diameter and the relative velocity between the particle and the air. This is shown in Figure 2a (in the original manuscript) in the shaded region.
* * *
- *A.8 : P4 L27: Can you speculate as to what is the physical (or numerical) meaning of this scaling relationship? Or is this a purely empirical finding?*

**Response A.8:** Following the work described in this manuscript, we explored this interesting relationship a bit further and we have reasons to believe that this quantity can actually be derived directly from equations (1) and (2) of the manuscript. This derivation is not yet complete and we leave it for future publications.
* * *
- **A.9 : P4 L31 Fig 1g-l: Please expand the negative range of $T_P$ - $T_{Air}$. There are environments where fohn events can bring dramatic changes of temperature up to 10C over only a few hours!**

**Response A.9:** A similar comment was raised Reviewer #2 and so we have increased the range in the updated figure to -5 K to 5 K. Figure 1(g-l) in the previous manuscript are Figure 3(a-f) in the revised manuscript.
* * *
- **A.10 : P5 L20: Look at parameterization**

**Response A.10:** This has been updated in the revised manuscript.
* * *
- **A.11 : P5 L26: It is unclear to me how using this stationary flow is fundamentally different from your steady state model. Was the LES used because it is a more sophisticated framework in which to calculate these fluxes? Besides the evolution of friction velocity is Figure S2, I am afraid I have missed the point of using such a complex tool to solve some PDEs.**

**Response A.11:** There are two principal reasons for using the LES. Firstly, we wanted to find out about the residence time of typical saltating ice grains. This information is not available in literature and so we decided to perform LES of a turbulent channel flow with a erodible snow surface as the lower "wall" of the flow. The surface acts as a source or sink of particles with simple stochastic models to account for different entrainment and deposition processes. The transport of particles is modeled by solving the equations of motion for each particle individually once the particle is eroded and is air-borne. The LES methodology for aeolian transport is well established and has been validated in the past. We realize that we have not cited past works in this section and have rectified this oversight.

The second motivation is in fact directly related to a previous comment by the reviewer (A.5). Unlike Experiment I and II, the air surrounding the particle ( thinking from the frame of reference of the particle ) is continuously evolving with different wind speed,temperature and humidity values. How do the two different approaches for computing sublimation (TM and NUM) compare in this scenario with complete feedback between air and ice grains ? This is question we answer in Section 3 using LES.

Within the LES context, by stationary turbulent flow, we intended to say that the logarithmic profile of the velocity is achieved and the time-averaged turbulent statistics ( or Reynolds averaged statistics ) are horizontally homogeneous and steady and vary only in the vertical direction. The wall-bounded channel flow that we simulate still has a significant shear in the vertical direction (as expected in the wall-bounded shear flows).

The LES also allows for simulating vertical gradients of temperature and humidity as would exist in nature. The vertical mixing of these scalars allows the sublimation of saltating ice grains to continue as dry air from aloft is continuously mixing downwards into the saltation layer. A detailed analysis of the heat and moisture budgets in the presence of the saltating ice grains will be presented in a future publication.

However, this and other comments have led us to believe that we have perhaps not motivated the use of LES sufficiently in the submitted manuscript, or described the LES in sufficient detail. Even though we go into great detail about the LES and the setup in the supplementary material, we expand the section 3.1 in the revised manuscript.

Additionally we submit a movie (Supplementary Video M1) illustration of the simulation we perform to make it clear the kind of LES we have performed.

––––––––––––––––––––––––

- ***A.12 : P5 L32: Why was this not discussed in the previous experiments?***

  **Response A.12:** The initial condition for particle was indeed discussed in the previous experiments but this was perhaps not clear due to lack of proper notation ( no mention of $T_{p,IC}$ in Section 2 for example). In the revised Section 2, we explicitly state that we are imposing $T_{p,IC}$ in the experiments in Section 2 as well.

––––––––––––––––––––––––

- ***A.13 : P6 L3: Please stop calling this realistic saltation of snow. The parameterizations and assumptions necessary to run this numerical model make this statement misleading. Please rephrase as LES simulation of saltation or something similar.***

  **Response A.13:** We have removed the word "realistic" from the sentence. Adding "LES simulation of saltation" does not seem to be appropriate as Experiemt III is purely about using LES. The entire sentence now reads as follows:
  *The principle aims of Experiment III are to firstly quantify particle residence times (PRT) and their dependence on wind speeds and relative humidities and secondly, compute the differences in the heat and mass output between the NUM and the TM approaches during saltation of snow with complete feedback between the air and the particles.*

––––––––––––––––––––––––

- ***A.14 : P6 L5: These varying friction velocities are referred to as "low medium and high wind speeds" in line 33. What wind speeds were necessary for these values? Friction velocities do a poor job of representing turbulence in even subtly complex terrain, and as saltation is a drag-driven process, at the very least, mean wind speeds should be included in the manuscript, and extensive time series of turbulence statistics (Turbulence Intensity, TKE, shear stress, etc.) in the supplementary material. As this research is conducted to benefit those that models in natural settings, and those natural setting will be much more turbulent than the LES, and that turbulence is what is driving the ventilation rates, more information about the model is needed.***

  **Response A.14:** We thank the reviewer to bringing to our attention this fact. The TO BE DONE !

- **A.15 : P6 L5: Where is "the surface" defined for this stress calculation? And how is that surface defined? How can that be reconciled with the fact there is windpumping into the snow pack? Or is this a Reynolds-stress-based value?**

  **Response A.15:** In terms of the forces that the surface imparts to the overlying fluid, the surface is treated as a rough wall. The roughness is parameterized using a roughness length $(z_0) = 10^{-5}m$. This approach does not account for the windpumping into the snowpack. We mention this in the revised text.
* * *
- **A.16 : P6 L15: Why a different range of temperatures than Section 2?**

  **Response A.16:** The range of temperatures in Experiment II has now been increased to -5K to +5K.
* * *
- **A.17 : Fig 2a: Redo the plots so it is clear what is happening. I cannot understand anything from 200 $\mu$m to 1000 $\mu$m. The diameter plot markers appear somewhat logarithmically. Try plotting with a logx scale? And why do the residence time measurements become more sparse at smaller particle sizes? Please redo the symbols as they are confusing and inconsistent, or eliminate them altogether.**

  **Response A.17:** We thank the reviewer for pointing out lack of clarity in Figure 2. This figure is the most essential part of the paper and thus, it is extremely important for us to make sure that it is well understood by our readers.

    – We have now restricted the figure to 600 $\mu$m. There are not enough particles larger than 600 $\mu$m and thus the statistics are noisy.
    – The x-axis of the figure is indeed logarithmic. We have added this information in the figure's caption.
    – The markers were added only for differentiating and labeling the different trend-lines. Not all data-points have been marked.
    – As mentioned in the submitted manuscript at P5 L21-22: *The snow surface consists of particles with a log-normal size distribution with a mean particle diameter of 200 $\mu$m and standard-deviation of 100 $\mu$m..* The particle size distribution (PSD) imposed on the surface comes from previous studies of modeling of saltation of snow. The PSD constrains the particle diameters that are air-borne and undergo transport. Also, we use a continuous spectrum and thus, when calculating statistics of mean and median residence times, we use a fixed bin size of 25 microns. As Figure 2a has a logarithmic x-axis, the measurements appear to be sparse at the lower range of the diameters.
* * *
- **A.18 : P6 L23: There is no dependence on $\sigma$?**

  **Response A.18:** No, the relaxation time $\tau_{relaxation}$ does not depend on $\sigma_*$. This is one of the remarkable results of Section 2 and we now make this point more explicitly in the revised manuscript.

- **A.19 : P6 L29: Please elaborate why the values of mass loss are wrong. It appears in Fig 1c,f that the cumulative errors go to zero over time. Why is this no longer the case with LES?**

**Response A.19:** Once again, we feel that we could have perhaps done a better job in explaining the relationship between Experiments I/II and Experiments III/IV.

The cumulative errors in Figure 1c,f *tend* towards zero but for a solitary ice grain. In the LES, a particle, original resting at the surface, is made air-borne (either due to aerodynamic entrainment or splashing ), makes multiple hops across the snow surface, where is rebounds from the surface, and ultimately comes to rest, i.e, it impacts the surface and does not rebound. In the LES, there are many thousands of particles that go through this cycle during the course of the simulation. Since models parameterizing the erosion and deposition of the particles are stochastic, particles in saltating have a range of hops, distance traveled and residence times. Additionally are a range of particle diameters present in the flow. We track the residence time of each particle, and calculate statistics of mean and median residence time as a function of diameter.

It is found that the smaller grains ( with diameters less than 150 microns ) have "on average" residence times that are longer than the relaxation time. Thus for these particles only, the cumulative errors averaged over multiple particles, will indeed *tend* to zero. The LES also have particles ( with diameters greater than 225 microns ) that have residence times "on average" larger than the plausible values of the relaxation time. Thus, for only these particles, the cumulative errors of mass and heat output will **not** go to zero. Summing all these errors for all the particles in the flow, the total error is non-zero. In fact Figure 3 shows precisely this error and it is found to range from 28% to as high as 40 % in Experiment III.

Thus, the LES are not performed for a single ice grain, with different simulations for different particle diameters. The LES is performed of a turbulent channel flow with an erodible snow surface consisting of a distribution of particle diameters at the lower wall. The ice grains enter and exit the flow at the surface according to models governing the erosion and deposition mechanisms. The supplementary movie M1 will aid in making this point clear.

- **A.20 : P7 L1: Please rephrase "larger-scale turbulence statistics." It unclear to me how any "larger-scale turbulence" can be represented in a 6×6×6 meter box. Is this not an increase in mean windspeed?**

**Response A.20:** By "larger-scale turbulence statistics", we meant to say that the dynamics of the heavy particles to be invariant to different flow speeds. We simplify the statement as follows:
*This means that the dynamics of the heavier particles are unaffected by different wind speeds simulated in Experiment III.*

- **A.21 : P7 L3-8: This is a very interesting finding! This suppression of vertical motions and how the model responds should be elaborated on! A comparison of the vertical turbulence statistics amongst the experiments is necessary as**

*they all assume uniform initial air temperature (P6 L5 comment). How does vertical mixing in the LES deal with this over time? Logic would imply that this same suppression of vertical mixing could also be caused by a colder snow surface temperature and increased stability. Why have you disregarded particle surface temperature in your PRT experiments? Would this effect be found if a temperature gradient as found in nature were present, or would the numerical effect be overwhelmed by the near-surface temperature gradients? As it stands, this statement cannot stand alone and the conclusion needs more development and supporting data/plots would be very beneficial.*

**Response A.21:** We agree with the reviewer that this is indeed an interesting finding. We have added an entire section in the supplementary material providing a preliminary analysis of this phenomenon by showing the vertical profiles of the vertical buoyancy flux. However, as we explain in the "comments_to_all.pdf" document, this is an ancillary result that is not directly related to the core message of the paper. The role of buoyancy in mediating aeloian transport is a very interesting and as-of-yet unexplored topic. We are in fact working on this topic currently and hope to present results focusing on this topic in the coming months.

Coming to the additional questions posed by the reviewer, we answer them as follows:

– Accounting for surface temperature is not likely to have a major impact on the stability of the atmosphere in strong snow drift events that we are considering. Whether the snow is sublimating on the surface, or during transport, both processes are going to result in stable stratification of the atmosphere. However, the amount of sublimation and the resulting cooling is much more from the particles in air, in comparison to those lying on the surface. In our simulation, where we have fully developed saltation/snow transport, the effect of the sublimation, and stability due to surface sublimation is likely to be negligible in comparison to the corresponding effect emerging from particles in the air. Note that we have stably stratified air in our simulations as well. Just that the stability emerges due to sublimation of particles in the air and not on the surface. We agree that in intermittent snow transport conditions, the surface boundary condition will become important. This is a matter for further exploration.

– This effect would indeed be found if there is a temperature gradient present. Note that only the initial condition for temperature is fixed at 263.15 K. The temperature in the LES evolves with time and the atmosphere does become stably stratified.

– We stress again the fact that this, although an interesting result, is only ancillary to the core message of the paper and we stress upon this point more in the concluding section of the paper.
* * *
- *A.22 : P7 L3-8 These are very small particles, can they be considered in "Suspension?" Obviously, there is a full spectrum of motions, but approximately where have other researches been separating saltation from suspension on Fig 2a? This would be very informative as the paper by nature is a saltation study.*

**Response A.22:** We present results only for particles that saltate. There are indeed a few particles in "suspension", i.e, particles that once leaving the surface, never deposit during the course of the simulation. But the number of such particles is an order of magnitude lower than those that saltate. Residence times are thus computed only for particles that leave and return to the surface.

- **A.23 : P7 L12: Very exciting finding!**

  **Response A.23:** We agree !
* * *
- **A.24 : P7 L18: What is field scale?**

  **Response A.24:** We have removed this phrase in the revised manuscript.
* * *
- **A.25 : P7 L21: Can anything be said about the low end of the friction velocity domain where intermittent transport dominates? Would TM over or underestimate in that case?**

  **Response A.25:** No, intermittent transport is a very interesting phenomenon where a lot more research is required to simulate it properly. The initial friction velocities are chosen such that we have "fully-developed" saltation. Having said that, the TM would still underestimate the mass lost by the solid ice phase due to sublimation but the underestimation will be lower than those found in Experiment III.
* * *
- **A.26 : Fig 3: Where have the particle diameters gone? What distribution of sizes are you using?**

  **Response A.26:** The particle size distribution (PSD) is imposed as described on P5 L21-22: *The snow surface consists of particles with a log-normal size distribution with a mean particle diameter of 200 μm and standard-deviation of 100 μm.*. As mentioned earlier, we have now added a figure with the PSD in the revised manuscript.

  Fig 3 shows the "total" mass lost due to sublimation - from all the particles that have undergone sublimation during the simulation.
* * *
- **A.27 : P8 L22: Not perturbations.**

  **Response A.27:** We have replaced "temperature perturbations" with "temperature differences" in the revised manuscript.
* * *
**B: Technical Concerns**

- **B.1 : P1 L7 Please specify: snowpack surface temperature, snow particle surface temperature?**

  **Response B.1:** In the revised manuscript, the temperature of the snowpack surface temperature and the air flow is specified ( as 263.15 K).
* * *
- **B.2 : EQ3 What is "d"? $d_p$?**

  **Response B.2:** Yes, it is indeed $d_p$. This error has been corrected in the revised manuscript.
* * *
- **B.3 : P2 L24 "Saturation $\sigma_*$ ..." ? Do you mean sigma is saturation?**

  **Response B.3:** In fact, $\sigma_*$ is the rate of saturation ( or saturation-rate). The corresponding line is corrected in the revised manuscript as:
  $saturation\text{-}rate\ (\sigma_*) = \rho_{w,\infty}/\rho_s\,(T_{a,\infty})$.
* * *
- **B.4 : P2 L27: Add space after sentence end.**

  **Response B.4:** The corresponding line has been corrected in the revised manuscript.
* * *
- **B.5 : P5 L12: "an erodible"**

  **Response B.5:** Appropriate corrections have been made in the revised manuscript.

---

## Author Comment (AC2) · 31 Aug 2018

**Response to reviewers:**
**On the suitability of the Thorpe-Mason model for calculating sublimation of saltating snow**

Varun Sharma, Francesco Comola and Michael Lehning

August 31, 2018

**A note to all reviewers**

Please see the additional document found in the "comments_to_all.pdf" file

**Response to Reviewer # 2**

**Opening Remarks:**
We thank Reviewer #2 for his/her critique of the submitted manuscript. We have updated the manuscript based on the advice received and provide in the following section, a point-by-point response to the questions posed and the clarifications sought.

Based on Reviewer #2's comments, we have added an entire section in the supplementary material providing vertical profiles of some mean and turbulent quantities for some of the LESs performed.
* * *
**A: Concerns in the main text**

- *A1 : P.2, L.5: Perhaps it will be a good idea to refer the previous sublimation simulations in suspension. e.g., Xiao et al., 2000, An intercomparison among four models of blowing snow, Boundary-Layer Meteorology, 97, 109-135.*

  **Response A.1:** We have now added a host of references in this section. The updated sentences read as follows:
  *This is true for both field studies (Mann et al., 2000), where sublimation losses are calculated using measurements, usually at the height of O(1 m), and in mesoscale modeling studies (Xiao et al.,2000; Dry and Yau, 2002; Groot Zwaaftink et al., 2011; Vionnet et al., 2014),*
* * *
- *A.2 : P2., L.9: "recent studies using high-resolution large-eddy simulations" is the reference really use LES? I could not confirm in that paper.*

**Response A.2:** We re-checked the reference and it is indeed correct that Dai and Huang, 2014 did not use LES but rather a RANS type simulation. We have edited the text to read as follows:
*However, recent studies using high-resolution steady-flow, Reynolds-averaged Navier-Stokes (RANS) type simulations (Dai and Huang, 2014) claim that sublimation losses in the saltation layer are not negligible,*
* * *
- **A.3 : P.2, L.24: "saturation $\sigma_*$" does it mean rate of saturation?**

  **Response A.3:** We agree with the view that confusion that using the word "saturation" may cause. We have edited all references to $\sigma_*$ as *saturation-rate*.
* * *
- **A.4 : P.2, L.25: "Groot Zwaaftink et al. 2011, 2014 should be Groot Zwaaftink et al. 2011". In Groot Zwaaftink et al. (2014), mass loss due to sublimation are neglected in the calculation.**

  **Response A.4:** This is indeed true and we have corrected this mistake.
* * *
- **A.5 : P.2, L.27: "Vionnet et al., 2014).In" should be "Vionnet et al., 2014). In" (please add a space).**

  **Response A.5:** This edit has been made in the revised manuscript.
* * *
- **A.6 : P.3, L.9: The units should be given in Roman type, I think.**

  **Response A.6:** All units have to updated to Roman type in the revised manuscript as well as the supplementary material.
* * *
- **A.7 : P.4, L.3: Is the first-order scheme sufficient in the calculation?**

  **Response A.7:** Yes, given the extremely small time-step of 50 microseconds, the first-order scheme is indeed sufficient. We mention the time-step in the revised version of manuscript as follows:
  *For the NUM approach, Eqs. (1) and (2) are solved in a coupled manner using a simple first-order finite-differencing scheme for time-stepping with a time-step of 50 $\mu$s.*
* * *
- **A.8 : P.5, L.12: "a erodible" should be "an erodible".**

  **Response A.8:** This has been corrected in the revised version of the manuscript.

  ―――――――――――――――――

- **A.9 : P.6, L.15: Do you have any specific reason for the different temperature conditions (-5¡dT¡5 in EX. IV, -2.5¡dT¡2.5 in EX. II).**

  **Response A.9:** We have updated the temperature range in EX. II to be between -5 and 5K. The corresponding figures as well as the associated text have been updated as well.

  ―――――――――――――――――

- **A.10 : P.7, L.7-8.: "low initial saturation results in more sublimation and cooling near the surface, resulting in suppression of vertical motions." This is interesting indeed. Could you show the modification of the vertical profiles (temperature, sublimation rate, wind speed etc.) to illustrate these processes?**

  **Response A.10:** Please refer to the section " A note to all reviewers" with regards to this question.

  ―――――――――――――――――

- **A.11 : P.7, L.18: "at a field scale" specifically, during realistic saltation of snow?**

  **Response A.11:** We have modified this line to read as follows in the revised manuscript: *We can directly assess the implications of differences in grain-scale sublimation between the two approaches on total mass loss rates during saltation at larger spatial scales as simulated using LES in Experiments III and IV.*

  ―――――――――――――――――

- **A.12 : P.8, L.2: "here)." should be "here." (an unnecessary parenthesis).**

  **Response A.12:** We have corrected this in the revised manuscript.

  ―――――――――――――――――

**B: Concerns in the supplementary material**

- **B.1 : P.2, L.26 (S9): "ln" should be given in Roman type.**

  **Response B.1:** We have corrected this in the revised supplement.

  ―――――――――――――――――

- **B.2 : P.3, L.27: I think parenthesis is missing around the reference.**

  **Response B.2:** We have corrected this - by removing the reference to Groot Zwaaftink et al., 2014.
* * *
- **B.3 : P.3, L.29: $\sigma_{d_p}$ is undefined, I think.**

  **Response B.3:** This line is now as follows:
  *characterized by the mean, $\langle d_p \rangle$ and standard deviation, $\sigma_{d_p}$*
* * *
- **B.4 : P.4, L.2-4: Could you include the relevant references (Clifton and Lehning (2008) ?).**

  **Response B.4:** We have added this reference in the revised supplement.
* * *
- **B.5 : P.4, L.16-17: Is the rebound angle the same as that for sand? I think Kok and Renno (2009) obtained the results for sand. Do you hypothesis the angle is similar to sand saltation?**

  **Response B.5:** This is indeed true, but a previous work (Nalpanis et al. 1993) made wind tunnel experiments with different granular media, including and and snow and found that the saltation geometries, ejection and rebound angles are invariant. We cite this work in the revised supplement.
* * *
- **B.6 : P.4, L.19-20: "dislodge additional additional particles" should be "dislodge additional particles".**

  **Response B.6:** This has been corrected in the updated supplement.
* * *
- **B.7 : P.7, L.23: Is the first-order scheme adequate for the computation in this study?**

  **Response B.7:** Yes - considering that we use an extremely small time-step of 50 microseconds, it is adequate. Additionally, a higher order method would require additional memory.
* * *
- **B.8 : P.8. "Time step" All the elements (fluid, particle, and scalars) have the same timestep?**

  **Response B.8:** Yes - all equations are progressed in time using the same time-step.
* * *
- **B.9 : P.9, L.5: "It shows that that once" should be "It shows that once"?**

  **Response B.9:** This has been corrected in the updated manuscript.

---

## Author Comment (AC3) · 31 Aug 2018

**Response to reviewers:**
**On the suitability of the Thorpe-Mason model for calculating sublimation of saltating snow**

Varun Sharma, Francesco Comola and Michael Lehning

August 31, 2018

**A note to all reviewers**

Apart from addressing the points raised by of each of the two reviewers individually, we felt that it might be pertinent to write to both the reviewers together especially in view of the fact that the most outstanding issue raised by both is essentially the same - the need to present more data.

At the outset, we would like to thank the reviewers for mostly positive remarks regarding the quality of the submitted manuscript and the core message of the paper, i.e., questioning the well-established, Thorpe-Mason (TM) model for computing sublimation of snow particles in the atmosphere. We are further grateful to their detailed comments regarding the study and clarifications sought. We feel that this has genuinely improved the manuscript and help us convey our ideas more clearly to the prospective reader.

Coming to the common concern raised by both the reviewers, both the reviewers felt that we need to present more data, especially from the large-eddy simulations (LES) in Section 3 and used for Experiments III and IV. As would be apparent from the length of the submitted manuscript, we intended to be extremely focused on the core message - challenging the Thorpe-Mason model - and presented only those results that we felt were *most* relevant to support our result and make it clear to the community that this is a worthwhile challenge to the existing models and estimates for sublimation. We write in the originally submitted quite explicitly that the large-eddy simulations are, in the context of the current study, used for two purposely only; (a) to find the residence time of saltating particles and (b) see if the results from the extremely idealized grain-scale simulations of sublimation are relevant during saltation. These two aspects are covered by Figures 2 and 3 in the original manuscript.

We agree with Reviewer #1 that we need to atleast state the flow speed in the different $u_*$ cases and we have added them in the revised manuscript. However, we humbly state that presenting vertical profiles of various mean and turbulent quantities, although quite interesting, would make the article lose its focus and move the discussion into areas that are far from the core message of the paper. This is especially true because we have 30 simulations in total. Given the immense size of the data-set, presenting it in any which way would certainly make the paper unwieldy, without perhaps adding much to core message of the paper.

Are the vertical profiles of different variables interesting ? Certainly. Indeed, we are currently working towards a manuscript analyzing the LES results using mean and turbulent kinetic energy, heat and moisture budgets. The present round of reviews has in fact motivated us even further

about the importance of such an analysis for the community. However, the depth of the analysis necessitates a separate paper. In view of our opinion on this issue we state the following points:

- It seems to us that for both the reviewers, the interest in analyzing vertical profiles from the LES simulations was triggered by a hypothesis presented in the paper to explain the role of increased sublimation and the associated cooling and stabilization of the saltation layer in lowering the residence time of lighter (smaller) particles as opposed to the heavier (larger) particles. This hypothesis is indeed confirmed in our data and can be explained by looking at the total vertical buoyancy flux. However, presenting this quantity would require presenting a host of other associated quantities. Most importantly, this point by itself, although quite interesting and a major motivation for a lot of our current work, is not of principal interest for the core message of the paper. We have remolded the text in the revised manuscript accordingly and touched upon this point in the concluding section of the paper.

- If the reviewers feel quite strongly that this data is absolutely necessary in the current paper, we hope that it is indeed agreeable to the reviewers that it is sufficient to add it to the supplement (as suggested by Reviewer #1). We attempted to add it to the main portion of the manuscript but it was really hard to maintain the flow of the paper and not distract the reader into issues far from the core message of the paper. Thus, the reviewers will find, in what follows, the new additional analysis that we hope can be added to supplement.
* * *
REPRODUCED FROM THE REVISED SUPPLEMENT

**S3.1 Vertical profiles of mean and turbulent quantities**

In this section, vertical profiles extracted from the large-eddy simulations in Experiment III are shown to provide additional context for the simulations performed. Note that a detailed analysis of the vertical profiles is out of the scope of the present study and will be presented in a future publication. All the profiles presented below are time-averaged as well as averaged in the horizontal (periodic) directions.

In Fig. 1, the velocity magnitude for the low (UL), medium (UM) and high wind (UH) cases are presented. Influence of initial relative humidity (the RL, RM and RH variants) was not found to be important and thus not presented. Before the snow surface is allowed to be eroded, a fully developed channel flow is allowed to developed. This can be seen in Fig. S2 in the previous section. This also implies the formation of the logarithmic velocity profile as can be seen in Fig. 1. Once the snow surface is allowed to erode, snow transport begins and the particles in the flow cause enhanced drag in the flow. This causes the velocity profile to change with an overall deceleration of the flow. The wind speeds before saltation begins at a height of 1 m above the surface are 11 m/s, 16.33 m/s and 21.86 m/s for the UL, UM and UH cases respectively. Once the snow transport is fully-developed ( i.e., when the total mass of snow in the air is constant in Fig. S1 ), the corresponding wind speeds have reduced to 8.771 m/s (-20%), 11.34 m/s (-30%) and 12.98 m/s (-40%) respectively for the three cases.

The snow drift density is the mass of snow per unit volume present in the air and is shown for the UL-, UM- and UH- cases. Once again, the RL, RM and RH invariants of each of these cases are not found to be significantly different from each other and are thus not presented. Two time points chosen lie during the transient period where increasing snow mass is being entrained into air (profile at 10 seconds) and during fully-developed or steady state snow transport (profile at 240 seconds). The profiles are qualitatively as well as quantitatively (order of magnitude comparison) similar to previous works (see, for example, Gordon et al. 2009). As expected, the amount of

[Figure]

Figure 1: Vertical profiles of mean velocity magnitude for the low (UL), medium (UM) and high wind (UH) cases in Experiment III. For each case, profiles before commencement of saltation and 240 seconds after saltation begins are shown.

[Figure]

Figure 2: Vertical profiles of snow drift density for the low (UL), medium (UM) and high wind (UH) cases in Experiment III. For each case, profiles 10 seconds and 240 seconds after commencement of saltation are shown.

mass in the air increases with increasing wind speed and is found to be concentrated in the lowest 10 centimetres of air above the surface.

Figure 3 presents an inter-comparison of the evolving thermodynamic state of the air computed using either the NUM or the TM approach, with subfigures a,b and c showing vertical profiles of temperature, specific humidity and relative humidity respectively. Only one of the nine cases in Experiment III, namely the UL-RL case is chosen for illustration. Recall that as per our LES setup, the only source or sink of heat and mass in the atmosphere is through interaction with the particles. First, let's focus on subfigure c which shows the relative humidity (R.H) profiles at 3 different times after saltation begins, along with the initial condition for R.H, which in the UL-RL case was fixed at 30% in the entire domain. As time progresses, the R.H in the air increases due to cooling as well as larger amounts of water vapor, both due to sublimation of particles aloft. The profiles on the extreme right of subfigure c, which are extracted 1000 seconds after the start of saltation are similar for both the NUM and TM approaches with the air is close to saturation in both the cases. In the profiles at earlier time-steps, the R.H is higher near the surface and decreases with height as expected. The near surface air reaches a high saturation-rate ( 90%) within 100 seconds after saltation begins, but it takes almost 900 seconds more to reach saturation. This can be explained by turbulent mixing which continuously supplies dry air from aloft to the near-surface region.

While Fig. 3c, shows qualitatively a similar behavior for both the NUM and TM approaches as far as R.H evolution is concerned, we have shown in the main text that the TM approach underestimates the mass flux due to sublimation as compared to the NUM approach (see Figure 4 in the main article). The reason for this is the difference in the total cooling of the air between the two cases. This can be observed in the temperature profiles in Fig. 3a. For the TM approach, the cooling is much stronger, with the final temperature being 260.3 K, 2.85 K lower than initial air temperature of 263.15 K. On the other hand, for the final air temperature for NUM approach is 262.4 K, almost 2 K warmer than the TM case. The dynamics of the evolution of air temperature are much more complicated in the NUM case due to the inter-play between the thermodynamics of the air as well as the particles. Further work is needed to establish proper thermodynamic constraints on the coupled air-particle system. Ultimately, the results in Experiment I and II show that even for a solitary ice-grain, the TM approach under-predicts the mass sublimated in comparison to the NUM approach for exactly the same environmental conditions. This is reflected in the profiles of specific humidity in subfigure c. The NUM approach, at each of the three time-steps chosen shows higher flux as compared to the TM approach.

Vertical profiles of streamwise $\left(\sqrt{\overline{u'u'}}\right)$, cross-stream $\left(\sqrt{\overline{v'v'}}\right)$ and vertical $\left(\sqrt{\overline{w'w'}}\right)$ velocity fluctuations are shown in Fig. 4 for the UL-RL case before and during saltation. The TKE is highest near the surface and decreases with distance from the surface. Interestingly, during snow transport, each of the TKE components show a decrease as compared to their respective value before snow transport, upto a height on approximately 2 m above the surface. Above this height, the TKE components actually show an increase.

In the final figure in this section, we compare profiles of vertical buoyancy fluxes $\left(\overline{w'b'} = \left(g/\langle\theta_v\rangle_{xy}\right)\overline{w'\theta'_v}\right)$ from three cases, UL-RL, UL-RM and UL-RH, from Experiment III. The three subfigures show profiles for three different times after beginning of saltation. The vertical buoyancy flux is an important quantity as it is a term of the budget equation for vertical velocity fluctuations $\left(\sqrt{\overline{w'w'}}\right)$. For each simulation case, the buoyancy flux decreases as time progresses. The UL-RL case is also found to have the largest magnitude of buoyancy flux close to the surface in each of the time-steps shown, followed by UL-RM and finally UL-RH, which has the least buoyancy flux amongst the three cases. Note that this is negative buoyancy flux and thus, in terms of vertical velocity fluctuations, the -RL, -RM and the -RH cases have increasing vertical fluctuations in that order. This could potentially explain the results in Fig. 3a in the main text, where the lighther particles show increasing residence times in the order -RL, -RM and -RH. Further exploration of role of

[Figure]

Figure 3: Intercomparison between the NUM (redlines) and TM (green lines) approaches for calculating sublimation of saltating snow in the UL-RL case in Experiment III. The magenta line is the initial condition for temperature, specific humidity and relative humidity. In all the subfigures, the solid, broken and dotted lines are profiles extracted 100, 240 and 1000 seconds after the commencement of saltation respectively.

[Figure]

Figure 4: Vertical profiles of the three different constituents of the turbulent kinetic energy before and during saltation (240 seconds after saltation begins) for the case UL-RL in Experiment III.

[Figure]

Figure 5: Vertical buoyancy fluxes for three cases, UL-RL, UL-RM, UL-RH at different times after the commencement of saltation.

buoyancy in affecting saltation dynamics is left for future work.

REPRODUCED FROM THE REVISED SUPPLEMENT

---

## Author Response (AR2)

**Response to reviewers:**
**On the suitability of the Thorpe-Mason model for calculating sublimation of saltating snow**

Varun Sharma, Francesco Comola and Michael Lehning

October 3, 2018

**1 Response to Reviewer # 1**

We thank Reviewer # 1 for his/her comments on the revised manuscript and our answers to the questions raised in the previous round of review. We address the remaining concerns below.

- **A.1 : Response A.14: It is unclear whether this is the complete answer to the comment or not. Please clarify if this is a typo.**

  **Response A.1:** The response to A.14 was indeed a typo. The question and the intended answer are reproduced below.

  *A.14 : P6 L5: These varying friction velocities are referred to as "low medium and high wind speeds" in line 33. What wind speeds were necessary for these values? Friction velocities do a poor job of representing turbulence in even subtly complex terrain, and as saltation is a drag-driven process, at the very least, mean wind speeds should be included in the manuscript, and extensive time series of turbulence statistics (Turbulence Intensity, TKE, shear stress, etc.) in the supplementary material. As this research is conducted to benefit those that models in natural settings, and those natural setting will be much more turbulent than the LES, and that turbulence is what is driving the ventilation rates, more information about the model is needed.*

  **Response A.14:** In the revised manuscript, wind speeds at the height of 1 m above the surface are mentioned for each the different $u_*$ cases. We agree that this information is valuable and deserves to be added. Some turbulent statistics are presented in the revised supplement.
* * *
- **A.2 : Response A.21: The authors responded to a question of the impact of surface sublimation on suppression of vertical motions. However, the question posed was actually on the potential presence of a temperature gradient in the atmosphere as could be found in a typical stable boundary layer in calm wind over snow. However, as is mentioned below, it is suggested that these comments on vertical suppression just be removed altogether. It is unclear whether it was intended to imply that there is no such temperature gradient in nature with turbulent mixing, or that such a temperature gradient only exists**

*if there is actively sublimating surface snow?*

**Response A.2:** We agree with the suggestion that comments on vertical suppression are removed from the main portion of the manuscript.
* * *
- **A.3 Author Response Page 4, First paragraph: with the air is so close Please correct.**

  **Response A.3:** Corrected!
* * *
- **A.4 Author Response Page 4, Last Paragraph lighther.**

  **Response A.4:** Corrected!
* * *
- **A.5 Page 5 Line 3: Please add citation that these values are typical for saltation.**

  **Response A.5:** Some citations have been added. Note that since the work of Thorpe and Mason, not many studies explicitly talk about Nusselt and Sherwood numbers for ice grains.
* * *
- **A.6 P6 L15-16: Do you mean or**

  **Response A.6:** Yes, the sentence has been corrected.
* * *
- **A.7 Section 3.1 The use of LES is very attractive to many people in hydrology and snow science, but the complexity of such an endeavor is often the limiting factor. Please explicitly cite the software used, be it in-house code or a modified commercial product? It is appreciated that the governing equations of motion are included, but for a model of this nature, a note on where the code was actually developed (or ideally, available) would be beneficial to the community. As well, please note in the main body of text that is LES of a channel flow.**

  **Response A.7:** The code has been developed completely in-house over a period of 20 years with multiple dissertations contributing to it's development and applications. We note this information in the text. Additionally, it is explicitly stated that the LES is indeed of a channel flow.

- **A.8 P8L22: to decelerate**

  **Response A.8:** Thank you, it stands corrected!
* * *
- **A.9 P10L7: Fix units**

  **Response A.9:** Thank you, it stands corrected!
* * *
- **A.10 P10L8: "Mean wind speeds." It may be stationary, but it is still a turbulent flow.**

  **Response A.10:** Thank you, the sentence is now modified.
* * *
- **A.11 P10L13-17: This doesnt seem to be supported by the supplement. It is not clear that there was vertical suppression because of sublimation, only vertical suppression during saltation. This claim seems like it would need further investigation, for example a comparison with saltation without sublimation to show the actual impact of the cooling. Obviously TKE drops, but that is insufficient to show SUPPRESSION of vertical motions by thermal effects. The buoyancy is a function of vertical motions, so that is a bit of a circular argument. I suggest removing comment of any causal relationship between sublimation and vertical motion suppression until the referenced future research is conducted.**

  **Response A.11:** Similar to the answer to point A.2, we remove any statements linking vertical motions and sublimation etc.

  With regards to our discussion here, we would like to state that the vertical profiles in the supplement are for simulations with the same dynamical forcing and thus, the number of particles in saltation. The only difference between the three cases is that they have different initial conditions of relative humidity. In other words, the principle difference between the three profiles is the varying amounts of sublimation. Given the initial condition of uniform relative humidity, the buoyancy comes only from the sublimation ( cooling of the air ) due to particles and can be thought of as a cause. The effect is the vertical motions - if we write the Reynolds averaged equation for the vertical velocity fluctuation, we can see that all other terms being equal, the vertical motions are directly related to the buoyancy forcing - thus, more sublimation, more negative buoyancy and thus reduced vertical motions.

  We thank the reviewer for his/her suggestion for simulations of saltation with and without sublimation. This will address the link between sublimation and residence times definitively. This will indeed be a part of our future efforts.

- **A.12 Figure 1: The 40% reduction in windspeed with the presence of saltation is substantial and should be commented on. There are numerous wind tunnel (and some field) studies addressing momentum extraction from the wind by particles in saltation. This 40% should be put in context of previous work to add credibility to an inherently complex model of a simple scenario to be found in nature.**

  **Response A.12:** Citations of some papers discussing the velocity reduction have been added. Note that for most studies, the wind velocities, whether in the field or in wind tunnels are either lower than our UM case.
* * *
- **A.13 It is claimed in the manuscript that turbulent mixing of different temperatures is responsible for the regeneration of warm dry air that then results in different rates of sublimation for LES. It would be illuminating to have a movie showing an animation of evidence of that process, as it is the heart of the conclusions presented here.**

  **Response A.13:** An additional supplementary movie M2 is now added to the supplementary material. In M2, we show the evolution of the relative humidity instead of simply temperature. It can be clearly seen in the video, how eddies transport air with low relative humidity from aloft closer to the surface.
* * *
**2 Response to Reviewer # 2**

We thank Reviewer # 2 for agreeing for the manuscript to proceed for publication. We once again thank the Reviewer for his/her comments and suggestions during the review process that has undoubtedly improved the quality of the paper.

[revised manuscript text omitted]